# QUANTIFYING ZERO-SHOT COORDINATION CAPABILITY WITH BEHAVIOR PREFERRING PARTNERS

## ABSTRACT

Zero-shot coordination (ZSC) is a new challenge focusing on generalizing learned coordination skills to unseen partners in cooperative Multi-Agent Reinforcement Learning. Existing methods train the ego agent with partners from pre-trained or evolving populations. The agent's ZSC capability is typically evaluated with a few evaluation partners, including humans and agents, and reported by mean returns. Current evaluation methods for ZSC capability still need improvement in constructing diverse evaluation partners and comprehensively measuring ZSC capability. In this paper, we aim to create a reliable, comprehensive, and efficient evaluation method for ZSC capability. We formally define the ideal 'diversity-complete' evaluation partners and propose the best response (BR) diversity, which is the population diversity of the BRs to the partners, to approximate the ideal evaluation partners. We propose an evaluation workflow including 'diversity-complete' evaluation partners construction and a multi-dimensional metric, the Best Response Proximity (BR-Prox) metric. BR-Prox quantifies the ZSC capability as the performance similarity to each evaluation partner's approximate best response, demonstrating generalization capability and improvement potential. We re-evaluate strong ZSC methods in the Overcooked environment using the proposed evaluation workflow. Surprisingly, the results in some of the most used layouts fail to distinguish the performance of different ZSC methods. Moreover, the evaluated ZSC methods lack the ability to produce enough diverse and high-performing training partners. Our proposed evaluation workflow calls for a change in how we efficiently evaluate ZSC methods as a supplement to human evaluation.

## 1 INTRODUCTION

Zero-shot Coordination (ZSC) is a new challenge in training an agent named ego agent to have the capability to coordinate with unseen partners in cooperative Multi-Agent Reinforcement Learning (MARL) (Hu et al., 2020). The prevailing population-based training (PBT) (Jaderberg et al., 2017) methods always train the ego agent to be the common best response (BR) to the pre-trained population and evaluate the ego agents with a number of evaluation partners including humans and agents (Strouse et al., 2021; Yu et al., 2023). It is imperative to recognize that the diversity within the pre-trained population is crucial for an ego agent to obtain ZSC capability (Zhao et al., 2023; Charakorn et al., 2023). Concurrently, the diversity within the pool of evaluation partners is pivotal in manifesting the ZSC capability effectively (McKee et al., 2022b). Since human evaluation is expensive and unrepeatable, human proxy agents(Carroll et al., 2019), trained agents (Li et al., 2023a; Strouse et al., 2021), random agents (Xue et al., 2022) and rule-based agents (Charakorn et al., 2023) are used as supplemental evaluation.

However, current evaluation methods exhibit deficiencies that make it difficult to assess the ZSC capability fully. Specifically, the widely used human proxy agents are not similar to human (Yu et al., 2023). Moreover, we observe that current training-based methods of constructing evaluation partners using different training hyper-parameters, such as seeds and learning rates (Strouse et al., 2021), inevitably lead to the similarity between the training and evaluation populations. As summarized in Table 1, such defects of the current evaluation partners indicate that we urgently need a new method to construct evaluation partners for a comprehensive ZSC capability evaluation.

The Best Response (BR) classes hypothesis (Lupu et al., 2021; Rahman et al., 2023) means that the ego agent with strong ZSC capability can emulate any policy in the set of BRs to testing-time unknown partners. Based on the hypothesis, we formalize the concepts describing an ideal set of

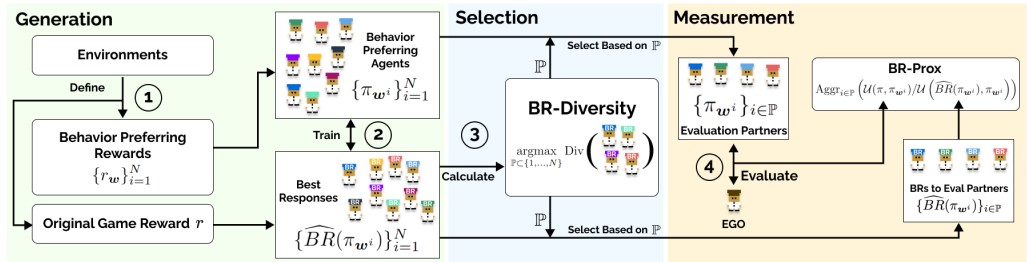

Figure 1: **Our Evaluation Workflow for ZSC.** 1) Generation: generating behavior preferring agents and their best responses; 2) Selection: selecting evaluation partners by maximizing the BR diversity; 3) Measurement: evaluating the ego agent with evaluation partners and compute BR-Prox. The detailed workflow is described in Section 4.

evaluation partners and highlight the concept of 'diversity-complete'. A set of evaluation partners with 'diversity-complete' BRs means the BRs to these evaluation partners also best respond to all other policies, thereby facilitating a comprehensive evaluation of ZSC capability. To approximate 'diversity-complete' evaluation partners, we establish the desiderata that evaluation partners need to be reasonable and diverse in skill styles and skill levels. We propose the Best Response Diversity (BR Diversity), which is the population diversity (Parker-Holder et al., 2020) of the partners' BRs to fulfill the desiderata. We justify that growing BR Diversity is more effective than growing the population diversity of the partners in selecting 'diversity-complete' partners from the given policies.

In this paper, we aim at a comprehensive and fair evaluation method, including the evaluation partners construction and the metric for ZSC capability measurement. As shown in Figure 1, we propose an evaluation workflow that includes generating a large number of evaluation partner candidates with diverse event-based behavior preferences, selecting the most representative ones as evaluation partners, and measuring the ZSC capability via our proposed BR-Prox metric. Our BR-Prox quantifies the ZSC capability as the performance similarity between the ego agent and approximate BRs to the evaluation partners. Realizing that the ego agent trained by ZSC methods aims to generalize learned skills to evaluation partners, BR-Prox can illustrate both the generalization capability and the improvement potential as a multi-dimensional metric. Using this evaluation workflow, we re-evaluate the ZSC methods in the Overcooked environment Carroll et al. (2019). The results show that the performance of current methods cannot be distinguished in the most used layouts, which indicates the effectiveness of our evaluation workflow in revealing the improvement potential. We also quantify the effects of population sizes on the population diversity, and our findings indicate that existing methods cannot produce enough diverse and high-performing partners for training.

To sum up, our main contributions are as follows: 1) To the best of our knowledge, we are the first to investigate the evaluation of ZSC capability and formally define the concepts of ideal evaluation partners; 2) We propose an evaluation workflow including 'diversity-complete' evaluation partners construction and the BR-Prox metric that multi-dimensionally measures the ZSC capability, in which BR Diversity, a novel population diversity metric, is proposed and justified; 3) We provide a benchmark of ZSC methods with a reliable and comprehensive evaluation.

## 2 RELATED WORKS

**ZSC Problem and Methods.** Most existing ZSC methods are developed based on Self-play (SP) (Tesauro, 1994; Yu et al., 2022; Wang et al., 2022), which learns conventions between players and produces agents that lack coordination with unseen partners (Carroll et al., 2019). Representative methods aim to reduce the overfitting of conventions by using randomization over the game structure to break the known symmetries (Hu et al., 2020) and designing diversity-based reward shaping for robust strategies (Lucas & Allen, 2022). The most prevalent ZSC solution is population-based training (PBT) (Jaderberg et al., 2017) like Population Play (PP) (Carroll et al., 2019), which aims to train an ego agent by allowing the ego agent to interact within a population to ensure that the ego agent encounters multiple partners during training. Fictitious co-play (FCP) (Strouse et al., 2021) proposes a widely used two-stage framework, which we named as *Co-Play Methods*, pre-training a population via self-play and then training the ego agent with the pre-trained population. Most of the co-play methods pay attention to training a diverse population in the first stage by

Table 1: Evaluation partners used in recent works under the Overcooked environment.

| Evaluation Methods | Utilized by | Defects |
|---|---|---|
| Human Players | Overcook-AI (Carroll et al., 2019); MEP (Zhao et al., 2023); FCP (Strouse et al., 2021); HSP (Yu et al., 2023); PECAN (Lou et al., 2023); HiPT (Loo et al., 2023); COLE (Li et al., 2023b) | Expensive, time-consuming and unrepeatable |
| Human Proxy Agents | Overcook-AI (Carroll et al., 2019); MAZE (Xue et al., 2022); FCP (Strouse et al., 2021); MEP (Zhao et al., 2023); HSP (Yu et al., 2023); PECAN (Lou et al., 2023); COLE (Li et al., 2023a)   HiPT (Loo et al., 2023); | Divergent from human behaviors and not diverse |
| Trained Self-play Agents | FCP (Strouse et al., 2021); MAZE (Xue et al., 2022); LIPO (Charakorn et al., 2023); HiPT (Loo et al., 2023) | Similar to the training populations |
| Trained Adaptable Agents | MAZE (Xue et al., 2022);  COLE (Li et al., 2023a) | Unfair in cross-play |
| Rule-based Specialist | HSP (Yu et al., 2023); LIPO (Charakorn et al., 2023) | Inextensible and expert-needed |
| Random Agents | FCP (Strouse et al., 2021); MAZE (Xue et al., 2022) | Lack of high-level skills |

using population entropy-shaped reward (Zhao et al., 2023), using hidden-utility reward functions that model human behaviors (Yu et al., 2023), and training incompatible agents (Charakorn et al., 2023). The contextual encoding method to identify the partners' policies has also been used to encourage self-play agents to discover diverse behaviors in the pre-training stage (Lou et al., 2023; Loo et al., 2023). The other method is *Evolution Methods*, in which the ego agent is trained with an evolving population of partners. Current evolution methods keep updating the partners pool during the training by promoting unique behaviors among the strategies (Lupu et al., 2021), co-evolving (Xue et al., 2022) and open-ended learning (Li et al., 2023a;b).

**ZSC Evaluation and Analysis.** Many researchers have focused on the ZSC evaluation and analysis in both Human-agent team and agent-agent team. McKee et al. (2022b) present the empirical studies to evaluate agents' generalization gap and propose the expected action variation metric to measure the population diversity. Knott et al. (2021) suggested that the average training or validation reward cannot represent the agents' robustness. Otherwise, some studies discussed the subjective evaluation of Human-AI Team performance but did not focus on ZSC capability (Siu et al., 2021; McKee et al., 2022a). Unlike the above studies, we focus on evaluating ZSC capability. The current ZSC capability evaluation methods usually use different evaluation partners to showcase the ego agent's ZSC capability, including human and agents, as summarized in Table 1. The defects and other details are discussed in Section 3.2 and Appendix C. Due to the defects listed in Table 1, previous evaluation methods cannot provide a comprehensive and fair evaluation. To overcome these shortages, our method can build partners with diverse behaviors which are unseen during the training process to ensure comprehensive evaluation of ZSC capabilities. And the BR-Prox we introduced enables a fair comparison of different algorithms' ZSC performance by quantifying their capability in terms of generalization capability and improvement potential. To the best of our knowledge, we are the first to investigate the construction of evaluation partners and the measurement of ZSC capability.

## 3 PROBLEM SETTING AND MOTIVATION

### 3.1 TWO-PLAYER MARKOV DECISION PROCESS

We model the ZSC problem as a two-player Markov Decision Process (Boutilier, 1996) denoted by $\mathcal{M} = <\mathcal{S}, \mathcal{A}^1, \mathcal{A}^2, \rho, \mathcal{P}, r, \gamma>$, where $\mathcal{S}$ is the state space, $\mathcal{A}^1$ and $\mathcal{A}^2$ are the action spaces for the two players, $\rho : \mathcal{S} \mapsto [0, 1]$ is the distribution of the initial state $s_0$, $\mathcal{P} : \mathcal{S} \times \mathcal{A}^1 \times \mathcal{A}^2 \times \mathcal{S} \mapsto [0, 1]$ is the dynamics function that indicates the transition probability, $r : \mathcal{S} \times \mathcal{A}^1 \times \mathcal{A}^2 \mapsto \mathbb{R}$ is the reward function, and $\gamma$ is a reward discount factor. For two policies $\pi^1$ and $\pi^2$, a joint action $\boldsymbol{a}_t$ is sampled from the joint policy $\pi^1(\cdot|s_t) \times \pi^2(\cdot|s_t)$ at each step $t$ given state $s_t \in \mathcal{S}$. With the expected discounted return denoted as $\mathcal{J}(\pi^1, \pi^2) = \mathbb{E}_{\tau \sim (\rho, \pi^1, \pi^2)} \left[ \sum_t \gamma^t r(s_t, \boldsymbol{a}_t) \right]$, we define the objective considering the position permutation as $\mathcal{U}(\pi^1, \pi^2) = \frac{1}{2}(\mathcal{J}(\pi^1, \pi^2) + \mathcal{J}(\pi^2, \pi^1))$. For convenience, we denote the Best Response (BR) of a policy $\pi$ as $BR(\pi) = \text{argmax}_{\pi'} \mathcal{U}(\pi', \pi)$. Let $\Pi_{\text{test}}$ be the set of potential unseen partners, and $\pi^e$ be the policy of the ego agent. The optimization objective of the ZSC problem can be represented as: $\max_{\pi^e} \mathbb{E}_{\pi^p \sim \mathbb{U}(\Pi_{\text{test}})} [\mathcal{U}(\pi^e, \pi^p)]$, where we assume partners are sampled uniformly. As we focus on population-based ZSC methods, we further

formalize the objective considering the construction of the training population:

$$\max_{\Pi_{\text{train}}, \mathcal{O}} \mathbb{E}_{\pi^P \sim \mathbb{U}(\Pi_{\text{test}})} \left[ \mathcal{U} \left( \mathcal{O}(\Pi_{\text{train}}), \pi^P \right) \right] , \tag{1}$$

where $\Pi_{\text{train}}$ is the population constructed during training and $\mathcal{O}$ is an approximate oracle function that computes the common best response for partners in $\Pi_{\text{train}}$. For instance, the oracle function can be defined to maximize the objective with $\mathbb{U}(\Pi_{\text{train}})$, i.e., $\mathcal{O}(\Pi_{\text{train}}) = \text{argmax}_{\pi} \mathbb{E}_{\pi^P \sim \mathbb{U}(\Pi_{\text{train}})} [\mathcal{U}(\pi, \pi^P)]$.

## 3.2 DESIDERATA FOR EVALUATION PARTNERS

Note that agents in ZSC problem encounter unknown partners in test-time, and thus the construction of the evaluation partners plays a significant role in evaluating ZSC capability. The BR classes hypothesis (Lupu et al., 2021; Rahman et al., 2023) supposes the common BR policy to a large set of partners is robust to a large number of unknown partners, which means the ego agent with strong ZSC capability can emulate any policy in the set of BRs to testing-time unknown partners. Therefore, an ideal evaluation method should expose the ego agent to evaluation partners with 'diversity-complete' BRs. We can intuitively define a 'diversity-complete' set of BRs as the set contains all possible BRs. However, the ego agent is trained to coordinate with human or other agents to accomplish the given tasks in the ZSC problem (Hu et al., 2020; Lucas & Allen, 2022; Lupu et al., 2021). Thus the agents that sabotage the game are not targeted partners. We reduce the BRs to those sabotaging agents by defining a 'diversity-complete' set of BRs in an MDP $\mathcal{M}$ as $\text{DCS}_{\text{BR}}(\mathcal{M})$:

$$\forall \pi' \in \Pi, (\mathcal{U}(\pi', BR(\pi')) \geq \eta(\mathcal{M}) \rightarrow BR(\pi') \in \text{DCS}_{\text{BR}}(\mathcal{M})) ,$$

where $\Pi$ is the policy space in $\mathcal{M}$, $\eta(\mathcal{M})$ is the threshold for identifying the sabotaging agents and depends on the environment. We remark that there is a similar concept in Rahman et al. (2023), while we outline the concept in the context of ZSC problem. Accordingly, we formally define the set of evaluation partners with 'diversity-complete' BRs as $\text{DCS}(\mathcal{M})$:

$$\left( \exists \text{DCS}'_{\text{BR}}(\mathcal{M}), \{BR(\pi)\}_{\pi \in \text{DCS}(\mathcal{M})} = \text{DCS}'_{\text{BR}}(\mathcal{M}) \right) \wedge \left( \forall \pi \in \text{DCS}(\mathcal{M}), \mathcal{U}(\pi, BR(\pi)) \geq \eta(\mathcal{M}) \right) ,$$

which means that BRs to the set of evaluation partners $\text{DCS}(\mathcal{M})$ is 'diversity-complete' and that $\text{DCS}(\mathcal{M})$ does not include sabotaging agents. We briefly describe such set of evaluation partners as 'diversity-complete'.

**Problems of previous evaluaton methods.** Currently, the researchers use different ways to construct the evaluation partners. We categorize the partner agents in Table 1 and provide more details in Appendix C. Considering the substantial expense and time consumption in human evaluation, current evaluation methods employ partners generated by training or rule-based methods. However, current methods still need to work on producing reasonable partners with diverse skill styles and levels. Specifically, Yu et al. (2023) highlighted that the most widely used human proxy agents for evaluation do not have human-like behaviors. The trained self-play agents used for evaluation may not be distinct from the training partners. As shown in Figure 2, we demonstrate that the populations trained with MEP under different training settings learn similar behaviors[1]. The cross-play evaluation method is flawed because evaluation partners overlap with training partners, and excluding the overlapped partners disrupts the control tests, which may have led to an unfair comparison. The manually built rule-based specialists are impractical to implement in complex environments. The random agents lack high-level performance to ensure the evaluation partners are diverse enough.

Therefore, we urgently need 'diversity-complete' evaluation partners to help evaluate the ZSC algorithm more efficiently. Unfortunately, we cannot precisely obtain a 'diversity-complete' evaluation partners set since whether the partners are 'diversity-complete' or not depends on the environment and the given task and may be intractable in complex environments. Instead, we can establish the desiderata of the evaluation partners to approach 'diversity-complete' evaluation partners:

**Reasonable.** We require evaluation partners to be reasonable in an informal sense. First, as explained above, partners should not minimize the episode return or have the potential to sabotage the game. Secondly, we require the partners not to be random since the performance with random behaviors and conventions does not contribute to the evaluation of ZSC capability. Therefore, we need to approximate the BRs to the generated partners to verify the partners' reasonableness.

---

[1] The details of this experiment can be found in Appendix B.3.

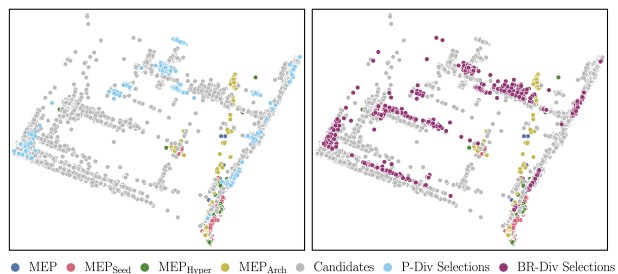
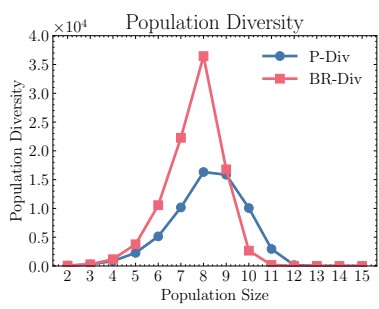

● MEP   ● MEP$_\text{Seed}$   ● MEP$_\text{Hyper}$   ● MEP$_\text{Arch}$   ● Candidates   ● P-Div Selections   ● BR-Div Selections

Figure 2: Visualization of the high-level behaviors of different self-play populations and our evaluation partner candidates. The evaluation partner selections are sampled according to partner diversity and BR diversity respectively.

Figure 3: Population diversity of subsets with different sizes. Near $0$ values mean that linearly correlated evaluation partners are included.

**Skill Style and Skill Level Diverse.** Previous studies have proposed methods for producing a variety of agents by optimizing the divergence in their low-level behaviors (Zhao et al., 2023; Lupu et al., 2021). Nevertheless, the number of possible low-level behaviors is frequently so large that it may be simple to maximize diversity objectives without learning qualitatively distinct policies. Moreover, locally different decisions do not necessarily exhibit distinct high-level behaviors. For a comprehensive evaluation, we require evaluation partners with diverse skill styles. Diverse skill levels are also essential when constructing evaluation partners. Fortunately, by selecting earlier checkpoints of evaluation partners, it is simple to acquire evaluation partners with diverse skill levels.

### 3.3 BEST RESPONSE PROXIMITY

Previous evaluation methods showcase the ego agents' ZSC capability by the mean episode returns across the evaluation partners. In contrast, multi-dimensional metrics should be adopted to measure the agents' capability to generalize the learned coordination skills to the unseen partners and the agents' improvement potential (Kirk et al., 2023). Given the evaluation partners and their approximate BRs, we are able to propose our Best Response Proximity (BR-Prox) metric that measures the performance similarity between the ego agent and the approximate BRs to the evaluation partners.

Formally, we define:
$$\text{BR-Prox}\left(\pi, \{\pi_{\boldsymbol{w}^i}\}_{i\in\mathbb{P}}\right) := \text{Aggr}_{i\in\mathbb{P}}\left(\mathcal{U}(\pi, \pi_{\boldsymbol{w}^i})/\mathcal{U}\left(\widehat{BR}(\pi_{\boldsymbol{w}^i}), \pi_{\boldsymbol{w}^i}\right)\right), \tag{2}$$
where Aggr means the aggregator across the evaluation partners, such as the most common 'mean' and 'median' aggregators. According to Agarwal et al. (2021), we use the inter-quartile mean as our aggregator, which is the mean of the middle 50% of data and is statistically reliable. Note that BR-Prox measures the ZSC capability by illustrating both the generalization capability and the improvement potential. Moreover, a single score can not provide a comprehensive profile of the performance distribution across the evaluation partners (Knott et al., 2021), as shown in Appendix B.3. To present an overall performance profile, we recommend reporting the results with the 95% confidence intervals (Nathaniel, 2021) and the inter-quartile ranges, e.g., the middle 50% of the disaggregated BR-Prox scores, as presented in Figure 4.

### 4 EVALUATION WORKFLOW

In this section, we first discuss constructing 'diversity-complete' evaluation partners using the best response diversity (BR Diversity), assuming access to the whole policy space. Then we propose our evaluation partners construction method, including evaluation partner candidates generation based on learning different behavior preferences and evaluation partners selection based on BR Diversity.

### 4.1 BEST RESPONSE DIVERSITY

Following Parker-Holder et al. (2020), we first define the **population diversity** of a population $\{\pi_i\}_{i=1}^M$ as the determinant of the similarity matrix: $\text{PD}(\{\pi_i\}_{i=1}^M) := \det(\boldsymbol{K})$, where $\boldsymbol{K}_{ij} = \boldsymbol{\theta}_i \cdot \boldsymbol{\theta}_j$ is the pair-wise similarity matrix of the population, and $\boldsymbol{\theta}_i$ is the behavior feature of policy $\pi_i$. The details of the behavioral feature are described in section 4.2.

Assuming the access to the whole policy space $\Pi$, one may intuitively construct 'diversity-complete' evaluate partners by repeating sampling subsets from the policy space and select the subset with the

maximum population diversity of the sampled partners, i.e., $\max_{\{\pi_i\}_{i=1}^M} \mathrm{PD}(\{\pi_i\}_{i=1}^M)$. However, maximizing the diversity amongst partners may not necessarily produce policies with specific conventions, particularly when a policy adheres to conventions that result in high returns when playing with the best-response policy to another evaluation partner (Rahman et al., 2023).

According to the definition of 'diversity-complete', we select 'diversity-complete' evaluation partners through maximizing the **Best Response Diversity** (BR Diversity, BR-Div), which is the population diversity of the approximate BRs to the selected partners. We define Best Response Diversity as BR-Div$(\{\pi_i\}_{i=1}^M) = \mathrm{PD}(\{\widehat{BR}(\pi_i)\}_{i=1}^M)$ and Partner Diversity (P-Div) as P-Div$(\{\pi_i\}_{i=1}^M) = \mathrm{PD}(\{\pi_i\}_{i=1}^M)$. To verify the effectiveness of constructing 'diversity-complete' policy subset from a given policy set, we present the maximum population diversity of the evaluation partner subsets with different sizes in the Overcooked environment (Carroll et al., 2019), as illustrated in Figure 3. We first pre-train a diverse pool of evaluation partner candidates and compare the subset selections according to P-Div and BR-Div. According to Figure 3, sampling based on BR-Div reaches higher population diversity before the population diversity diminishes, meaning that maximizing the best response diversity is more effective in constructing 'diversity-complete' evaluation partners than maximizing partners population diversity. We further demonstrate the effectiveness of maximizing BR diversity in Figure 2. The evaluation partners selected via maximizing BR-Div exhibit the most diverse behaviors while maximizing P-Div leads to inferior behavior discovering. Details of these experiments can be found in Appendix B.

## 4.2 Evaluation Partners Construction

We have justified the effectiveness of BR diversity in selecting a diverse subset from a set of policies. However, the whole policy space is generally unknown. In this section, we first generate candidates with diverse high-level behaviors through event-based reward shaping, which approximate the whole policy space and then select the most representative subset as the evaluation partners.

**Event-based Behavior Preferring Partners.** We expect the partners to have diverse and reasonable behaviors, while the the behavior space is intractably large. Inspired by Yu et al. (2023) in which human preferences are regarded to be event-centric and modeled as event-based reward functions, we apply the event-based reward shaping method (Abel et al., 2021) to encourage behavior discovering and generate behavior preferring partners. We formulate the reward space that approximates the human preferences by $\mathcal{R} = \{r_{\boldsymbol{w}} | r_{\boldsymbol{w}}(s_t, \boldsymbol{a}_t) = r + \phi(s_t, \boldsymbol{a}_t)^T \boldsymbol{w}, \boldsymbol{w} \in \mathbb{R}^m, \|\boldsymbol{w}\|_\infty \leq B_{\max}, \sum_i \mathbb{1}(\boldsymbol{w}_i \neq 0) \leq C_{\max}\}$, where $\boldsymbol{w}$ is an $m$-dimensional weight vector and $r_{\boldsymbol{w}}$ is the reward function encouraging behavior indicated by $\boldsymbol{w}$. The original game reward is added to prevent the behavior preferring agents from being sabotaging. $\phi : \mathcal{S} \times \mathcal{A}^1 \times \mathcal{A}^2 \mapsto \mathbb{R}^m$ embeds the event-based feature that counts the occurrences of each pre-defined event, i.e., $\boldsymbol{w}_j$ is the number of occurrences of the $j$-th event. $B_{\max}$ bounds the norm of $\boldsymbol{w}$ and $C_{\max}$ restricts the number of events which $\boldsymbol{w}$ concerns. Under these constraints, $\forall r \in \mathcal{R}$ indicate diverse behaviors but still encourage cooperation for solving the given tasks. Given such a reward space, partners with different behavior preferences could be generated and it is possible to derive diverse human behaviors. Given a specific $r_{\boldsymbol{w}}$, one player receives $r_{\boldsymbol{w}}$ and the other player receives the original game reward $r$. The procedure of the two players optimizing their objectives can be formulated as finding a Nash Equilibrium (NE) (Osborne & Rubinstein, 1994) in a two-player Markov Game (Van Der Wal, 1980). We can approximate an NE of such a two-player Markov Game by that two agents independently perform the Proximal Policy Optimization (PPO) (Schulman et al., 2017) algorithm since $r_{\boldsymbol{w}} \in \mathcal{R}$ still guides the behavior preferring agent to cooperate for solving the given task (Ding et al., 2022). After approximating the NE, we obtain the $\pi_{\boldsymbol{w}}$ which learns the behavior indicated by $\boldsymbol{w}$. We also obtain the approximate BR of $\pi_{\boldsymbol{w}}$, denoted as $\widehat{BR}(\pi_{\boldsymbol{w}})$. Line 1 to Line 3 in Algorithm 1 summarize the process of constructing agents that cover a set of diverse behaviors as the evaluation partner candidates by sampling reward functions $r_{\boldsymbol{w}}$ from $\mathcal{R}$.

**Evaluation Partners Selection.** The generated candidates may still be unbalanced distributed in cooperative conventions and behaviors as shown in Figure 2. To prevent the overestimation of the agents that coordinate well with those conventions and behaviors, we further select a representative subset of the candidates via maximizing the BR-Div. The policy behavior features used in the computation of BR-Div are environment-dependent and task-specific, pertaining to the skills under assessment. For simplicity, we count the occurrence of the pre-defined events of $\widehat{BR}(\pi_{\boldsymbol{w}^i})$ alongside the episodes generated by $\widehat{BR}(\pi_{\boldsymbol{w}^i})$ and $\pi_{\boldsymbol{w}^i}$ as the high-level behavior feature $\boldsymbol{\theta}_{\boldsymbol{w}^i} \in \mathbb{R}^m$ of

---

**Algorithm 1:** Behavior Preferring Evaluation Partners Construction

---

**Input:** Reward Space $\mathcal{R}$, Number of Candidates $N$, Number of Evaluation Partners $M$.

**Output:** Evaluation Partners $\{\pi_{\boldsymbol{w}^i}\}_{i\in\mathbb{P}}$ and Best Responses $\{\widehat{BR}(\pi_{\boldsymbol{w}^i})\}_{i\in\mathbb{P}}$.

1 **for** $i = 1, \ldots, N$ **do**

2      Sample $r_{\boldsymbol{w}^i}$ from $\mathcal{R}$.

3      Obtain $\pi_{\boldsymbol{w}^i}$ and $\widehat{BR}(\pi_{\boldsymbol{w}^i})$ by solving the two-player Markov Game.

4      Evaluate $\pi_{\boldsymbol{w}^i}$ with $\widehat{BR}(\pi_{\boldsymbol{w}^i})$ and embed $\widehat{BR}(\pi_{\boldsymbol{w}^i})$'s high-level behavior features
        $\mathbb{E}[\sum_{t=1}^T \phi(s_t, \boldsymbol{a}_t)]$ as $\boldsymbol{\theta}_{\boldsymbol{w}^i} \in \mathbb{R}^m$.

5 Compute the similarity matrix of $\{\widehat{BR}(\pi_{\boldsymbol{w}^i})\}_{i=1}^N$ as $\boldsymbol{K}$, where an element $\boldsymbol{K}_{ij} = \boldsymbol{\theta}_{\boldsymbol{w}^i} \cdot \boldsymbol{\theta}_{\boldsymbol{w}^j}$.

6 Sample a subset $\mathbb{S}$ of size $M$ as $\mathbb{S} = \operatorname{argmax}_{\{\pi_{\boldsymbol{w}^i}\}_{i\in\mathbb{P}}} \text{BR-Div}(\{\pi_{\boldsymbol{w}^i}\}_{i\in\mathbb{P}})$ via DPP sampling based on $\boldsymbol{K}$, where $\mathbb{P} \subset \{1, \ldots, N\}$.

7 Select checkpoints as $\{\dot{\pi}_{\boldsymbol{w}^i}\}_{i\in\mathbb{S}}$ with $\mathcal{U}(\widehat{BR}(\dot{\pi}_{\boldsymbol{w}^i}), \dot{\pi}_{\boldsymbol{w}^i}) \approx \mathcal{U}(\widehat{BR}(\pi_{\boldsymbol{w}^i}), \pi_{\boldsymbol{w}^i})/2, \ \forall i \in \mathbb{S}$.

---

$\widehat{BR}(\pi_{\boldsymbol{w}^i})$ for calculating the BR-Div, as summarized in Line 4 to Line 6 in Algorithm 1. Recall that the population diversity is defined as a determinant function. To search for the candidate subset of size $M$ with the maximum determinant, we apply the Determinantal Point Process (DPP) (Kulesza et al., 2012), which samples proportionally to the determinant of the candidate subsets. Due to the fact that candidate subsets are usually inexhaustible, we repeat the DPP sampling to search for the representative candidate subset with different subset initializations. Furthermore, we collect the earlier checkpoints of the selected evaluation partners to construct evaluation partners at different skill levels and additionally train the corresponding approximate BRs by PPO. The obtained evaluation partners approximate a 'diversity-complete' set of partners by meeting the desiderata since the selected evaluation partners are reasonable and diverse in skill-styles under the measurement of BR-Div and that the checkpoints at different skill levels are also included. Finally, these evaluation partners and approximate BRs are used to compute BR-Prox.

## 5   Overhauling Evaluation in Overcooked

In this section, we implement the strong ZSC methods in the Overcooked (Carroll et al., 2019) environment to quantify the performance of the current methods with different population sizes and compare the performance of the ZSC methods cooperating with the evaluation partners at different skill levels to show the effectiveness of our evaluation method and BR-Prox.

**Overcooked Environment**. We retain four layouts, including Asymmetric Advantages (Asymm. Adv.), Coordination Ring (Coord. Ring), Forced Coordination (Forced Coord.), and Counter Circuit (Counter Circ.) and add three new layouts which are Bothway Coordination (Bothway Coord.), Blocked Corridor (Blocked Corr.) and Asymmetric Coordination (Asymm. Coord.). The environment details and layout descriptions can be found Appendix A.

**Experiment Setup** We implement five strong methods, including FCP (Strouse et al., 2021), MEP (Zhao et al., 2023), TrajeDi (Lupu et al., 2021), COLE (Li et al., 2023a) and HSP (Yu et al., 2023), and additionally add self-play (SP) (Carroll et al., 2019) as a baseline[2]. For co-play methods, we collect the trained self-play agents and their earlier checkpoints as the training population following Strouse et al. (2021). For evolution methods, we initialize the population with a number of random agents. We adopt modern MARL techniques detailed in Appendix B.1, which help the implemented ZSC methods obtain higher episode returns different from results reported in original papers [3], as shown in Appendix B.3. We use a set of 14 events and $C_{\max} = 3$ to construct the event-based reward space, generate up to 194 candidates by traversing the reward space, and finally select up to 30 evaluation partners. More experiment setup details and full results can be found in Appendix B.

### 5.1   The Effectiveness of Our Evaluation Workflow

We re-evaluate five implemented methods in the Overcooked environment to show the effectiveness of our evaluation workflow. The main results and findings are as follows:

---

[2]We implement TrajeDi as a co-play method for a fair comparison and implement COLE without the bandit process for high parallelization.

[3]For example, self-play agents obtain nearly 500 episode returns in the Asymm. Adv. layout.

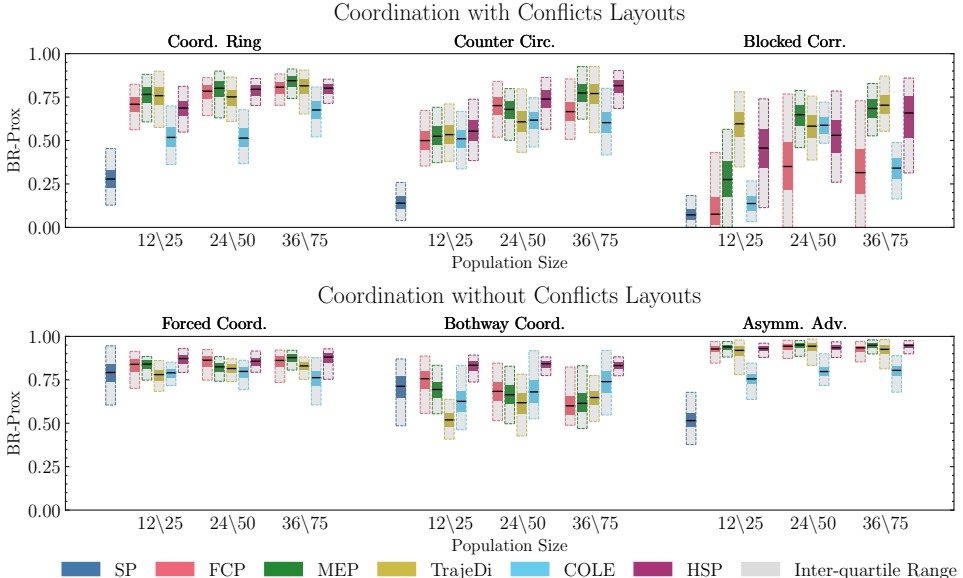

Figure 4: BR-Prox performance with 95% confidence intervals of ZSC methods with different population sizes in the Overcooked environment. '12\25', '24\50' and '36\75' mean that the co-play methods (FCP, MEP, TrajeDi and HSP) are trained with populations of 12, 24 and 36 sizes and that the evolution method (COLE) is trained with populations of 25, 50, 75 sizes.

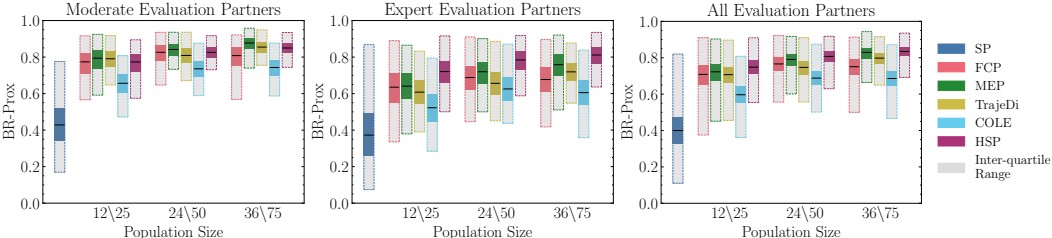

Figure 5: We compare the BR-Prox performance with 95% confidence intervals obtained with evaluation partners at different skill levels and aggregated from all layouts.

**Layouts Defects in Lack of Potential Skill Style Diversity.** Since BR-Prox reveals the improvement potential of ZSC methods, we find defects in the current environment layouts. As shown in Figure 4, we find that the most used layouts, including Forced Coord. and Asymm. Adv., cannot show the ZSC capability difference among the ZSC methods. We design the Bothway Coord. layout based on Forced Coord. to include more skill styles by encouraging the players to pass ingredients bidirectionally, which requires more coordination capability but still cannot tell the performance difference among the strong baselines. We believe that some current layouts cannot showcase the agent's ZSC capabilities due to the simple design and mechanisms. We used the inter-quartile range introduced in Section 3.3 to illustrate the variability of ZSC method performance in Figure 4. Interestingly, the self-play as a baseline method exhibits a high performance and a constricted inter-quartile range in the Forced Coord. and Bothway Coord. layouts, suggesting that the layouts lacking potential skill styles diversity allow agents to easily learn the majority of styles for interacting with unfamiliar partners. We find that resource sharing mechanism contributes the most to the amount of potential skill styles in a layout. To verify, we further name the layouts where the players need to share resources, such as the pot and the onion, and may have physically collision in movement as the 'coordination with conflicts' and the opposite as 'coordination without conflicts.' In the 'coordination with conflicts' layouts including Coord. Ring, Counter Circ. and Blocked Corr., a massive performance gap exists between methods. Such results indicate that there is still a considerable improvement potential for the ZSC problem in the scenarios where coordination leads to conflicts.

**Performance Gap with Evaluation Partners at Different Skill Levels.** Our evaluation workflow's effectiveness is also reflected in the performance gap of the methods with different skill levels evaluation partners. We investigate how the ZSC methods perform when facing unseen partners at

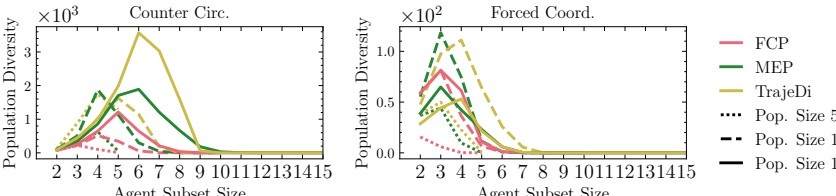

Figure 6: Effect of the population size on the population diversity. Near 0 values mean that linearly correlated evaluation partners are included.

different skill levels by regarding the evaluation partners with self-play performance less than the median, i.e., $\{\pi^P | \pi^P \in \Pi_{\text{eval}}, \mathcal{U}(\pi^P, \pi^P) \leq \text{Median}_{\Pi_{\pi^{P\prime} \in \text{eval}}} (\mathcal{U}(\pi^{P\prime}, \pi^{P\prime}))\}$, as moderate evaluation partners, and the left ones as expert evaluation partners. Results in Figure 5 reveal that ZSC methods have worse performance with the expert evaluation partners, including the lower BR-Prox values and larger variations. The deficiency may result from the ZSC methods failing to generate enough diverse and high-performing agents, which is elaborated in Section 5.2. Such a result indicates our evaluation partners at different skill levels show whether the ZSC methods can learn to cooperate with partners of different levels during the training process, which is what existing evaluation methods lack.

## 5.2 TRAINING POPULATION SIZE AND BR DIVERSITY

Besides, the proposed BR Diversity can also be an analysis tool for the effectiveness of ZSC methods generating training population. According to Figure 4 and 5, performance can be enhanced by increasing population size, provided that the population also increases in diversity, i.e., the population diversity is effectively enlarged by ZSC methods. Some ZSC methods lack an explicit mechanism to promote the population diversity, including FCP and COLE. Thus the performance of FCP and COLE is not benefited from increasing the population size from 24 to 36.

On the other hand, ZSC methods may sometimes fail to expand the population diversity. We provide a case study in the Counter Circ. and Forced Coord. layouts. Using the definition of population diversity in Section 4.2, we demonstrate the maximum diversity of the subsets of different sizes. For example, we compute the diversity of all the subset combinations with sizes ranging from 2 to 15 within a high-performing training population of MEP with a size of 15. The maximum diversities over the subset combinations of all subset sizes, respectively, are collected to plot the curve. Large diversities at the subset sizes close to the population size mean that the ZSC method effectively produces diverse agents. Results in Figure 6 illustrate that existing ZSC methods fail to effectively produce enough diverse high-performing agents, e.g., only eight diverse agents are produced after training a population of size 15 using MEP or TrajeDi. We can also explain the effects of population sizes on performance through the corresponding changes in population diversity: the increased population size leads to performance improvement when more diverse agents are produced (as in Counter Circ.) and vice versa (as in Forced Coord.).

## 6 CONCLUSION

In this paper, we investigate the evaluation of the zero-shot coordination capability. We are the first to formalize concepts describing the ideal evaluation partners for ZSC capability evaluation and highlight the shortages of existing evaluation partners. We design the BR-Prox metric to measure generalization capability and improvement potential. We justify the effectiveness of constructing 'diversity-complete' evaluation partners via maximizing our proposed BR diversity. Our evaluation workflow includes constructing the 'diversity-complete' evaluation partners by generating candidate agents with behavior preferences and their BRs, selecting the most representative ones by maximizing the BR diversity, and measuring the ego agent via BR-Prox. Five ZSC methods are re-evaluated in the Overcooked environment using the proposed evaluation workflow. Our evaluation workflow reveals that the most common layouts need more complexity to distinguish the performance of current ZSC methods. Results verify the effectiveness of our evaluation workflow in measuring the improvement potential. The BR diversity is further used to analyze the effectiveness of ZSC methods' training population generation, revealing their scarcity. For future work, we plan to reduce gap between the set of evaluation partners generated in our workflow and an ideal 'diversity-complete' set which mainly results from the events designed based on human knowledge. We also plan to introduce complex mechanisms into the Overcooked environment and apply our evaluation workflow to more environments for promoting the development of ZSC methods.

**Ethics Statement.** Our method and algorithm do not involve any adversarial attack, and will not endanger human security. All our experiments are performed in the simulation environment, which does not involve ethical and fair issues.

**Reproducibility Statement.** The source code of this paper is available at `https://anonymous.4open.science/r/ZSCBench`. We specify all the experiments implementation details including our implementation details of all baselines we evaluation and the experiments setup in Appendix B.1 The experiment additional results are in the Appendix B.3.

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

# Supplementary Material

## Table of Contents

## A  OVERCOOKED ENVIRONMENT

We re-evaluate in the Overcooked environment (Carroll et al., 2019). Overcooked is a simulation environment for reinforcement learning derived from the Overcooked!2 video game. The Overcooked environment features a two-player collaborative game structure with shared rewards, where each player assumes the role of a chef in a kitchen, working together to prepare and serve soup for a team reward. We retained 4 layouts including Asymmetric Advantages (Asymm. Adv.), Coordination Ring (Coord. Ring), Forced Coordination (Forced Coord.), and Counter Circuit (Counter Circ.) and added 3 new layouts: Bothway Coordination (Bothway Coord.), Blocked Corridor (Blocked Corr.) and Asymmetric Coordination (Asymm. Coord.). Visual representations of these layouts can be found in Figure 7.

**Forced Coordination.** The Forced Coordination environment is designed to necessitate cooperation between the two players, as they are situated in separate, non-overlapping sections of the kitchen. Furthermore, the available equipment is distributed between these two areas, with ingredients and plates located in the left section and pots and the serving area in the right section. Consequently, the players must work together and coordinate their actions to complete a recipe and earn rewards successfully.

**Counter Circuit.** The Counter Circuit layout features a ring-shaped kitchen with a central, elongated table and a circular path between the table and the operational area. In this configuration, pots, onions, plates, and serving spots are positioned in four distinct directions within the operational area. Although the layout does not explicitly require cooperation, players may find themselves obstructed by narrow aisles, prompting the need for coordination to maximize rewards. One example of an advanced technique players can learn is to place onions in the middle area for quick and efficient passing, thereby enhancing overall performance.

**Asymmetric Advantages.** In the Asymmetric Advantages layout, players are divided into two separate areas, but each player can independently complete the cooking process in their respective areas without cooperation. However, the asymmetrical arrangement of the left and right sides encourages collaboration to achieve higher rewards. Specifically, two pots are placed in the central area, accessible to both players. The areas for serving and ingredients, however, are completely distinct. The serving pot is placed near the middle on the left side and far from the middle on the right side, with the ingredients area arranged oppositely. Players can minimize their walking time and improve overall efficiency by learning how to collaborate effectively.

**Coordination Ring.** The Coordination Ring layout is another ring-shaped kitchen, similar to the Counter Circuit. However, this layout is considerably smaller than Counter Circuit, with a close arrangement that makes it easier for players to complete soups. The ingredients, serving area, and plates are all in the bottom left corner, while the two pots are in the top right. As a result, this layout allows more easily achieving high rewards.

**Bothway Coordination.** Compared to the Forced Coordination, Bothway Coordination enables both left and right agents to have access to onions and pots, giving them more policy space and cooperation forms, which decreases the long waiting time in Forced Coordination and enriches their policy diversity. Meanwhile, the plates and the serving spot are still placed to one side, thus the two players still need to cooperate to finish an order.

**Blocked Corridor.** In the Block Corridor layout, the most challenging part is the corridor which is the only connection between the left and right parts with the small throughput of one person in the middle. Both onions and plates are placed at the upper edge of U-Shape corridor, while pot and serving spot are placed at two bottom corner. If there is no cooperation at all, the onion need to be carried from upper left to lower right while the teammate needs to stay at the spare place at right side to avoid conflict. If we want to implement cooperation, there are a lot of options of spare counter, which provides many alternatives for how to cooperate. The agent needs to show its diversity and be able to response well to all possible behaviors of the player. Additionally, conflicting positions within small corridors is a challenge that needs to be addressed. Definitely, it is the most challenging layout of our setup.

**Asymmetric Coordination.** Modified from Asymmetric Advantage, this layout expands the map and changes the plates to be asymmetric. The first change expand the trajectory space. The second change make the right player have a shorter distance to pick a plate while the left player have a shorter to serve the soup, yielding a new cooperation form where right player pass the plate to left through the center counter.

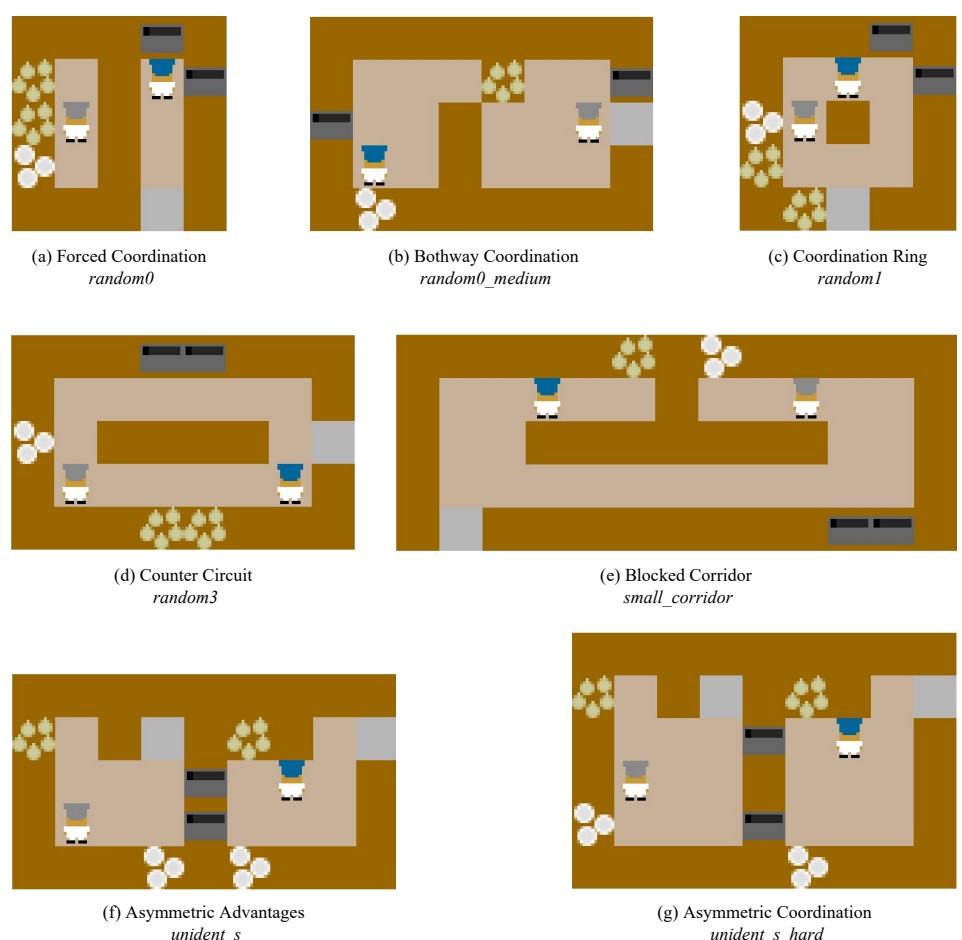

(a) Forced Coordination
*random0*

(b) Bothway Coordination
*random0_medium*

(c) Coordination Ring
*random1*

(d) Counter Circuit
*random3*

(e) Blocked Corridor
*small_corridor*

(f) Asymmetric Advantages
*unident_s*

(g) Asymmetric Coordination
*unident_s_hard*

Figure 7: Used layouts in Overcooked.

## B    Experiments Details and Additional Results

We implement 5 methods in Overcooked environment including self-play (Carroll et al., 2019), FCP (Strouse et al., 2021), MEP (Zhao et al., 2023), TrajeDi (Lupu et al., 2021) and COLE (Li et al., 2023a) based on the implementation of Yu et al. (2023) [4]. For each ZSC method and each layout, we evaluate the ZSC capability by the BR-Prox metric across 5 random seeds. For each seed, we evaluate the ego agent with 30 evaluation partners for the Asymm. Adv. and Asymm. Coord. layouts and 20 evaluation partners for other layouts, and for 50 episodes each partner. The code can be found in `https://anonymous.4open.science/r/ZSCBench`.

**Self-play.** Self-play (SP) is a general approach in reinforcement learning, where agents only learn through playing against themselves. While it can yield high returns during training, agents trained using this method often struggle to coordinate with diverse policies. We training $10,000,000$ steps for SP agents.

**FCP.** Fictitious Co-Play (FCP) is a two-stage training framework. In the first stage, it creates a diverse partner population through self-play agents pre-trained with different seeds and their previous checkpoints. In the second stage, it iteratively trains an FCP agent by having it play against sampled partners from the population. For the co-play methods including FCP, MEP and TrajeDi, we train $5e7$, $8e7$, $1e8$ steps for population sizes of 12, 24, 36 respectively.

**MEP.** Maximum Entropy Population-based training (MEP) is a variant of FCP. It adopts the maximum entropy as a population diversity bonus added to the task reward, which is used as the objective to train a maximum entropy population in the first stage. In the second stage, it trains an robust agent by prioritized sample agents from the population. We observe that $\beta$ for prioritized sampling should be small when the population size is large. Thus we use $\beta = 0.5$ in our experiments.

**TrajeDi.** Trajectory Diversity PBT (TrajeDi) aims to improve the policy diversity by adding a diversity measure to PBT losses. In details, it introduces the Jensen-Shannon divergence to the loss when training the population. We implement TrajeDi as a two-stage algorithm. We first train a population with the Jensen-Shannon divergence to encourage diversity and then train the ego agent with uniformly sampling the population. Due to the time consumption problem, we calculate the JSD by sampling the population instead of traversing the population.

**COLE.** Cooperative Open-ended Learning (COLE) constructs open-ended objectives in two-player cooperative games from the perspective of graph theory. With the objective functions calculated using the cooperative incompatibility distribution, it approximates the local best-preferred strategy to expand the population, which overcomes the cooperative incompatibility problem disclosed by other approaches. We implement the mete-solver using a reward-based ranking instead of the Shapley Value due to the time consumption. We train 50, 100 and 150 generations for population size of 25, 50 and 75 respectively and train 1,000,000 steps for a generation.

**HSP.** Hidden-utility Self-Play (HSP) constructs the training population is analogously to how we construct evaluation. HSP constructs a pool of behavior preferring agents using event-based rewards and select half of them by greedy-selection. The population is then used to train the ego agent with a mixture of behavior-preferring and MEP-trained partners. The main difference in population construction is that we use BR-Div to select evaluation companions and restrict the event-based reward space in order to promote reseaonable behavior.

### B.1    Important Implementation Details

**Parallel Partner Sampling**. When training the PPO algorithm, we sampling the episodes in which the ego agent plays with different partners in a batch, which makes the training framework more scalable.

**Centralized Critic**. Recent works have verified that a centralized critic function benefits the performance in fully cooperative games (Yu et al., 2022; Wang et al., 2022; Rashid et al., 2020).

---

[4] `https://github.com/samjia2000/HSP`

**Overcooked is an Truncated Infinite Game**. As emphasized in Gymnasium[5], Kostrikov & Raayai Ardakani (2020) and Pardo et al. (2018), wrong calculation of the truncated returns leads may break the MDP properties of the environments. We choose to discard the value function iteration from the truncated states.

**Available Actions**. We implement basic available action indications in the Overcooked environment, such as avoiding keeping hitting the counter and null interaction, to accelerate the exploration.

**Entropy Coefficients Decay**. To encourage discovering more high-performing coordination conventions, we choose to use large entropy coefficients and decay the entropy coefficients during training. The linear entropy coefficients decay mechanism is summaried in Table 2.

Table 2: Entropy coefficient schedulers.

| Method | Population Size | Entropy Coefficient Schedules | Entropy Coefficient Milestones |
|---|---|---|---|
| Co-play | 12 | 0.2 0.05 0.01 | $0\ 2.5e7\ 5e7$ |
| | 24 | 0.2 0.05 0.01 | $0\ 4e7\ 8e7$ |
| | 36 | 0.2 0.05 0.01 | $0\ 5e7\ 10e7$ |
| Evolution | 25 | 0.2 0.05 0.01 | $0\ 2.5e7\ 5e7$ |
| | 50 | 0.2 0.05 0.01 | $0\ 5e7\ 10e7$ |
| | 75 | 0.2 0.05 0.01 | $0\ 7.5e7\ 1.5e8$ |

**Population Size**. We choose the population size as 12, 24 and 36 for the co-play methods to demonstrate the effects of population size. While choose the population size as 25, 50, 75 for COLE since the evolution methods generate the ego agent end-to-end without pre-trained populations and thus require large populations to achieve better performance.

**Important Hyperparameters**. We use mostly the same hyperparamters as in Yu et al. (2023), except for the mentioned details such as the entropy coefficients.

**Event-based Reward Space Design and Policy Behavior Feature**. We design a set of events and their corresponding range of weights, as summarized in Table 3. Using $B_{\max} = 20$ and $C_{\max} = 3$, we generate up to 194 candidates and select up to 30 evaluation partners. The generated candidates are excluded if they cannot complete a delivery when cooperating with their BRs. The behavior feature of a policy is embedded as the occurrence of these events during the episodes.

| Events | Weights |
|---|---|
| Put an onion or a dish or a soup onto the counter | 0 |
| Pickup an onion or a dish or a soup from the counter | 0 |
| Pickup an onion from the onion dispenser | -20,0,10 |
| Pickup a dish from the dish dispenser | -20,0,10 |
| Pickup a soup | -20,0,5,10 |
| Place an ingredient into the pot | -20,0,3,10 |
| Deliver a soup | -20,0 |
| Stay | -0.1,0,0.1 |
| Movement | 0 |
| Order Reward | 0.1,1 |

Table 3: Designed events and weights used in Overcooked.

## B.2 VISUALIZATION OF BEHAVIOR PREFERRING PARTNERS

Figure 8 and Figure 9 show the heatmap of the evaluation partners' high-level behaviors in Coordination Ring and Asymmetric Coordination.

---

[5]`https://gymnasium.farama.org/tutorials/gymnasium_basics/handling_time_limits`.

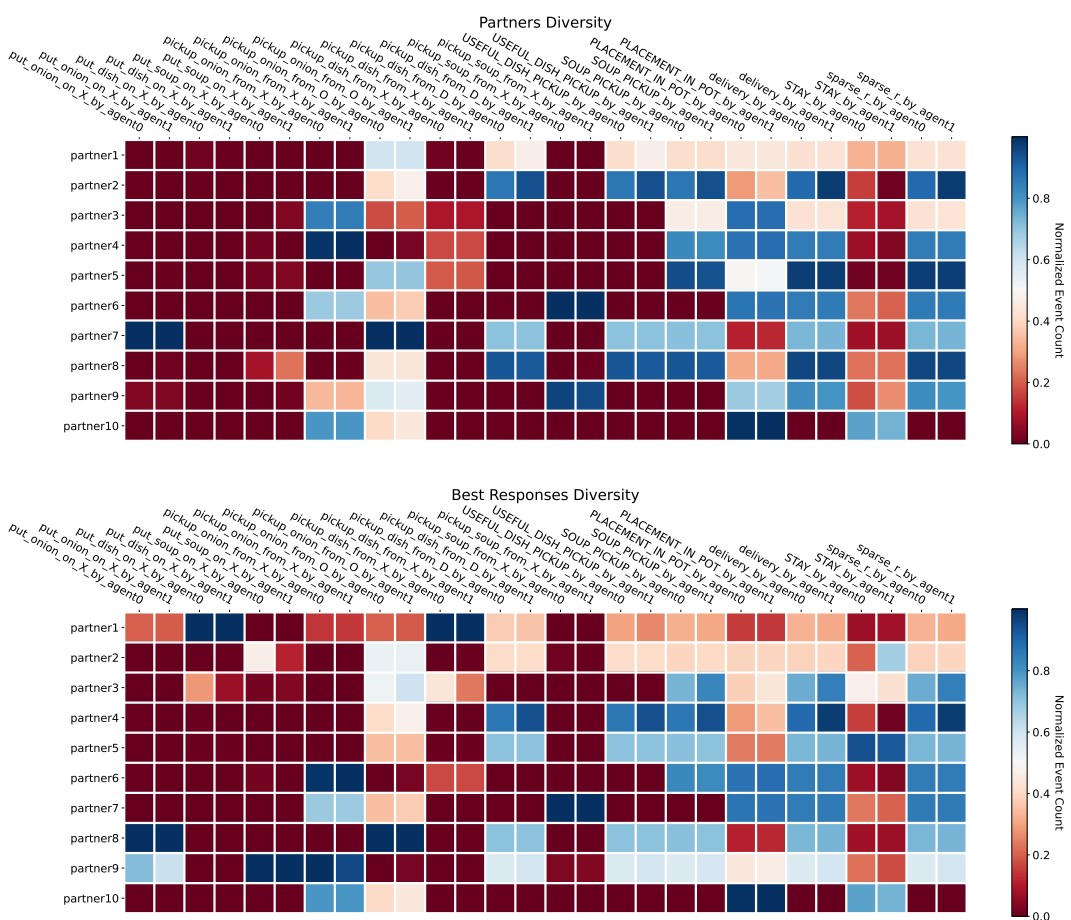

Figure 8: Heatmap of the evaluation partners' high-level behaviors of the Coord. Ring scenario in the Overcooked Environment. The BR-based Diversity maximization produces evaluation partners that use the counter more frequently.

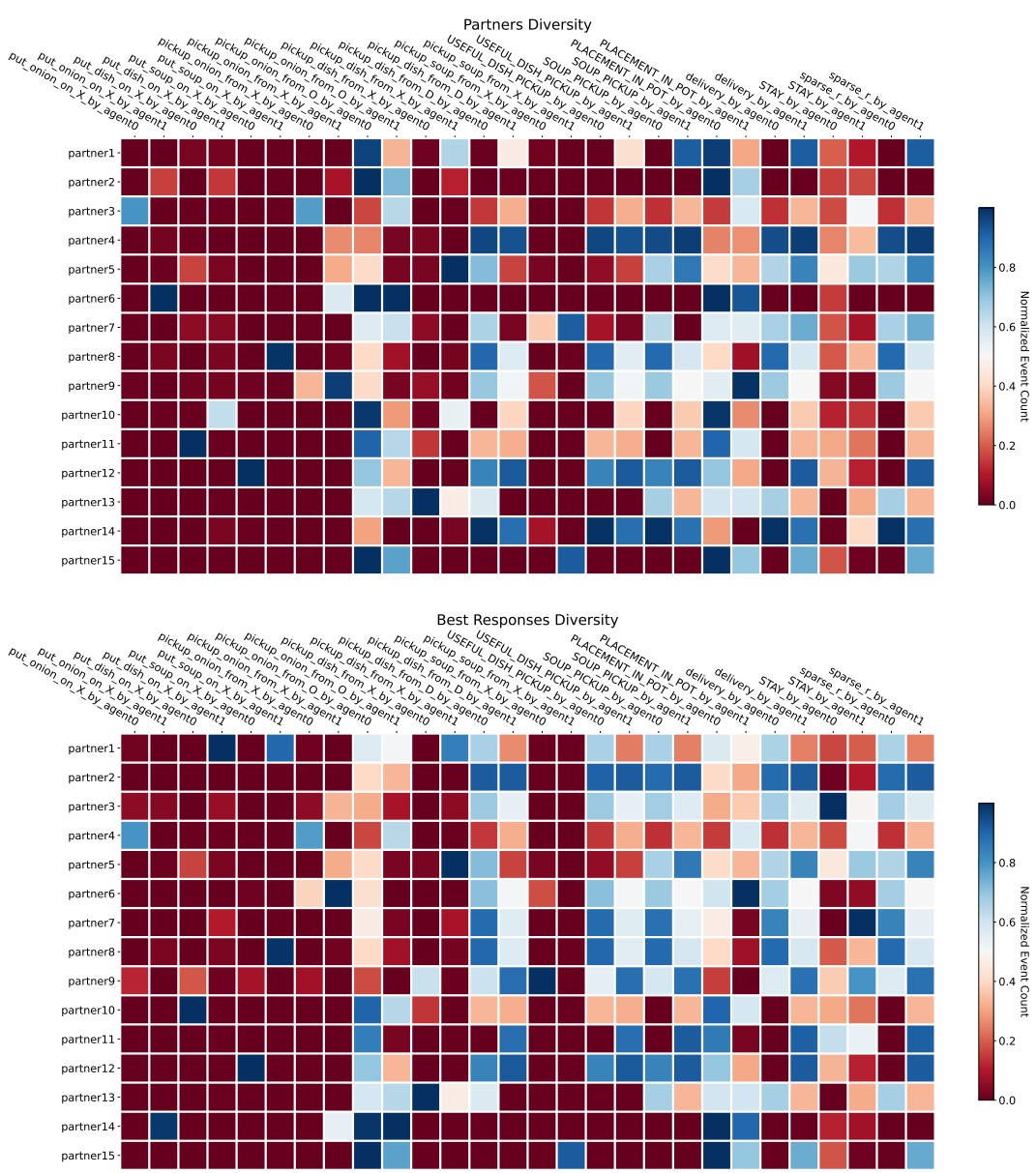

Figure 9: Heatmap of the high-level behaviors of the Asymm. Coord. scenario in the Overcooked Environment. The BR-based Diversity maximization produces evaluation partners that use the counter more frequently and deliver the soup in both sides.

### B.3 ADDITIONAL RESULTS

In Figure 2, we visualize the statistic data of the high-level events (Section 4.2) collected in Overcooked through principal components analysis (Dunteman, 1989). As is shown in the visualization result, populations trained using the MEP (Zhao et al., 2023) method differ in random initialization seeds, hyper-parameters and network architectures but learn similar behaviors.

Table 4: Percentage of Rank

| %         Rank
Algo | 1 | 2 | 3 | 4 | 5 | 6 |
|---|---|---|---|---|---|---|
| SP | 0.0 | 5.56 | 11.11 | 5.56 | 5.56 | 72.22 |
| FCP | 5.56 | 16.67 | 44.44 | 5.56 | 22.22 | 5.56 |
| MEP | 38.89 | 22.22 | 22.22 | 5.56 | 11.11 | 0.0 |
| TrajDi | 11.11 | 16.67 | 16.67 | 33.33 | 5.56 | 16.67 |
| COLE | 0.0 | 11.11 | 0.0 | 27.78 | 55.56 | 5.56 |
| HSP | 44.44 | 27.78 | 5.56 | 22.22 | 0.0 | 0.0 |

Figure 10 and Figure 11 show the performance of ZSC methods in all the 7 layouts. The black line marked on each bar is the interquartile mean of the data.

Table 4 summarizes the performance rank under BR-Prox with 3 different population sizes.

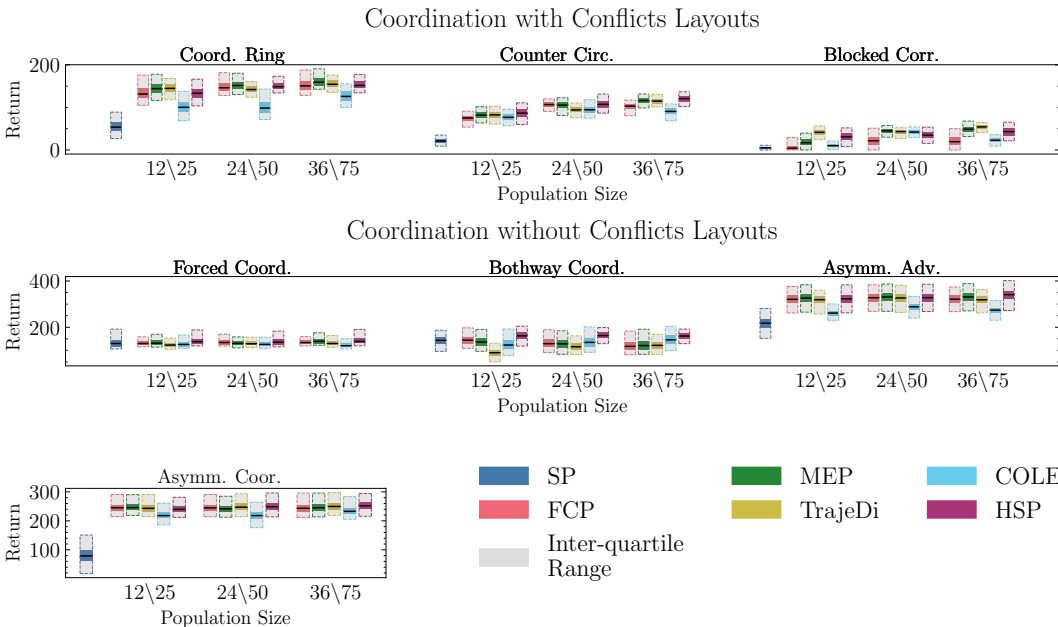

Figure 10: Episode return performance with 95% confidence intervals of ZSC methods with different population sizes in the Overcooked environment.

Figure 12 shows the diaggregated BR-Prox performance distribution with different population size in all the 7 layouts.

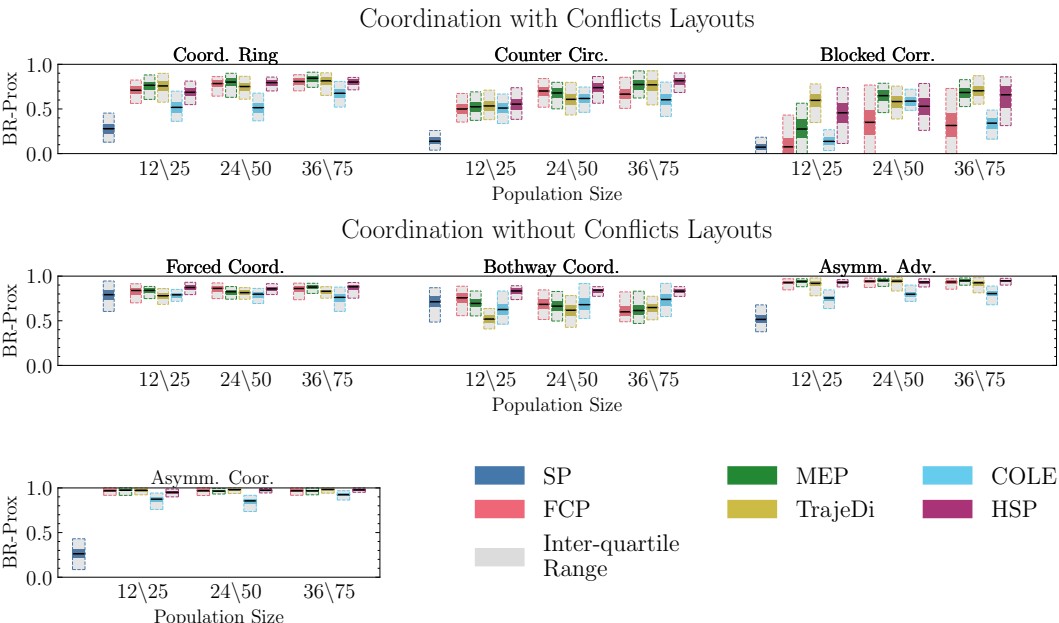

Figure 11: BR-Prox performance with 95% confidence intervals of ZSC methods with different population sizes in the Overcooked environment.

## C  CURRENT EVALUATION METHODS

We categorize the partner agents involved in the evaluation of these algorithms, referred to as evaluation partners, and offer a comprehensive analysis of the limitations associated with these partners.

**Human Players.** The human player in ZSC problem is a 'perfect' candidate for evaluation because human are strictly qualify as unseen partners. Many works present a human evaluation as a main contribution (Carroll et al., 2019; Strouse et al., 2021; Zhao et al., 2023; Yu et al., 2023; Li et al., 2023b; Lou et al., 2023; Loo et al., 2023). The most challenge is that during the training, a long-cycle human evaluation is not repeatable and cannot be replicated in large numbers to advance algorithm iterations. We need a more efficient evaluation method as a supplement for human evaluation.

**Human Proxy Agents.** The most widely used evaluation partners is human proxy agents proposed by Carroll et al. (2019). The human proxy agents were trained by imitation learning method with a human datasets, which aims to represent human behaviors and human diversity (Carroll et al., 2019). And (Strouse et al., 2021) used a similar way to construct a human proxy agents pool for evaluation. The availability and cost of human data are constrained. Yu et al. (2023) highlighted that these human proxy agents in overcooked environment do not account for human behaviors, which shows that using human proxy agents does not represent the diversity from humans and does not validate the ZSC capabilities of the algorithm and fully compare various methods.

**Trained Self-play Agents.** The trained self-play agents are also a widely used partners for ZSC capability evaluation (Strouse et al., 2021; Xue et al., 2022; Charakorn et al., 2023; Loo et al., 2023). However, through our experiment that comparing the similarities between two SP agents pools using different seeds (refer to Figure 2), we find that the diverse SP pool for evaluation which constructed using same algorithm in pre-training stage is similar to the pre-trained population in co-play methods.

**Trained Adapted Agents.** The evaluation via cross-play with other trained adapted agents is inevitable to have a part of evaluation that is tested on algorithm's own training set (even including their own ego) (Charakorn et al., 2023; Xue et al., 2022; Li et al., 2023a). We revisit again the core challenge of the ZSC problem: collaborating with an unseen partner. The own training set even including own ego may leads to a higher performance and they are not unseen partners. Therefore, the performance of cross-play does not completely reflect the capabilities of ZSC, and may cause the

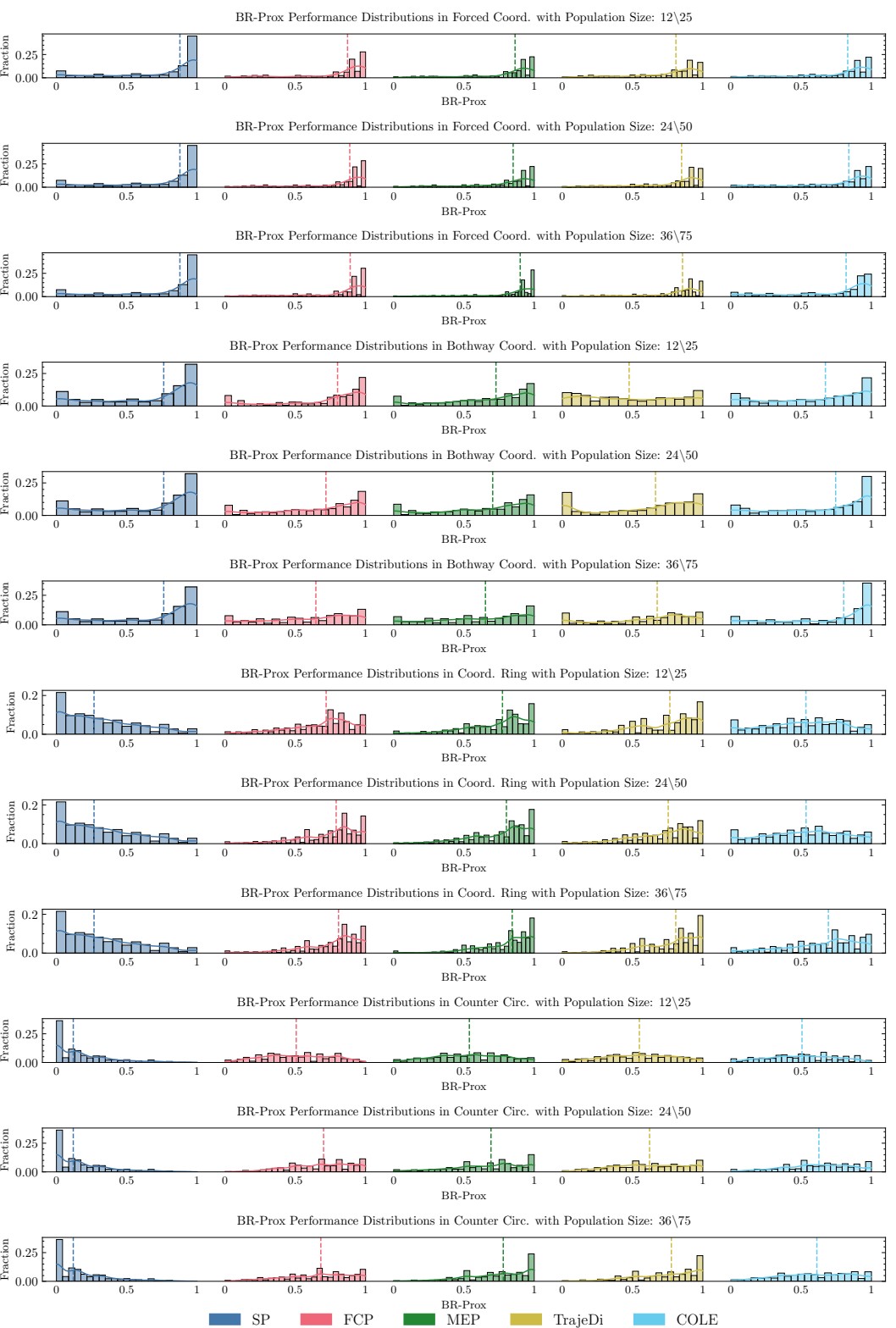

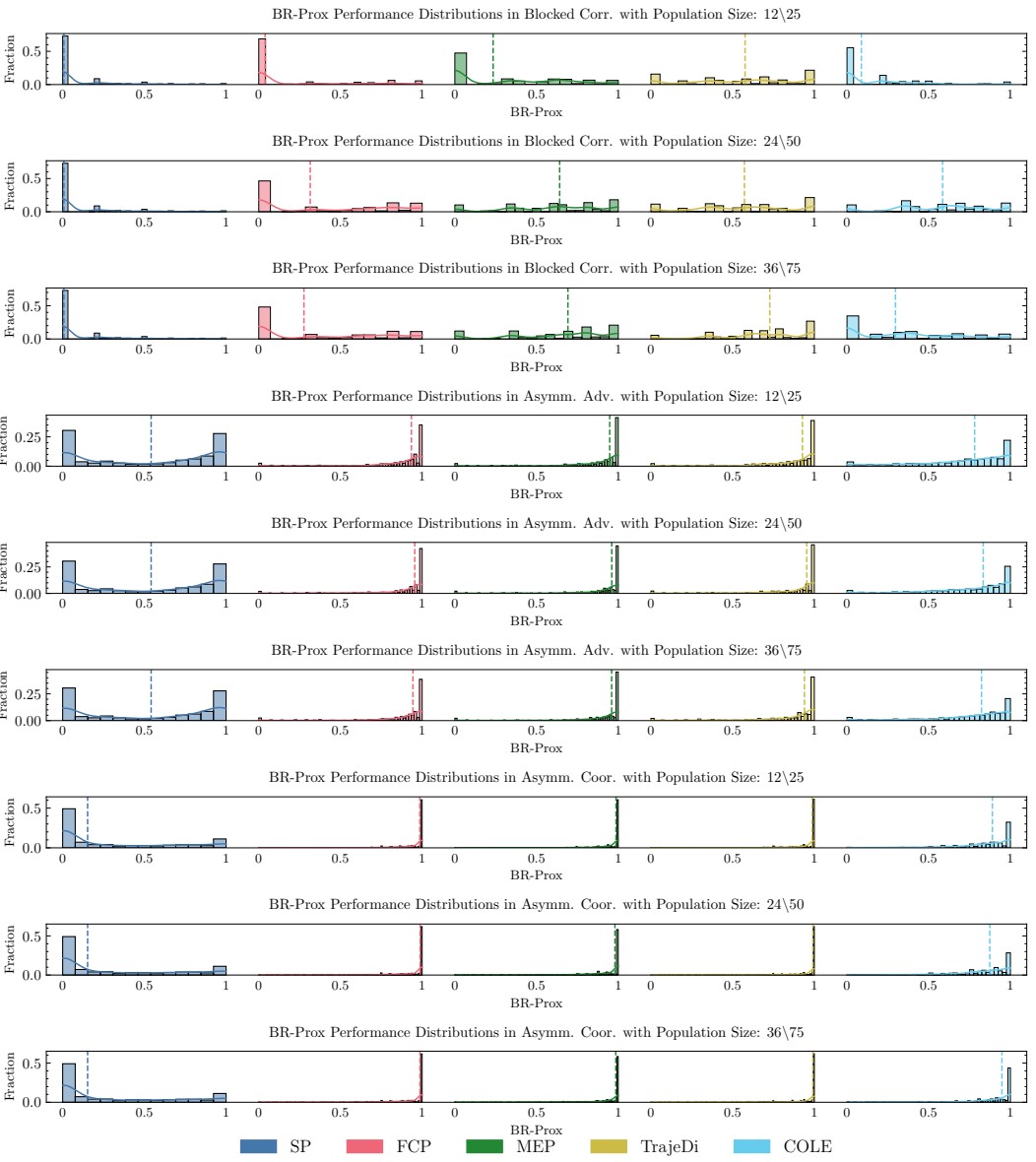

Figure 12: Distributions of 5 Algorithms with 3 Population Sizes in 7 Layouts.

performance to be falsely improved. And if excluding the performance from own training population to avoid the seen partners, the cross-play evaluation leads to a potential unfairness.

**Rule-based Specialist.** As a controllable method, using rule-based agents to evaluate the ZSC capability is also been used by some studies (Yu et al., 2023; Charakorn et al., 2023). The most problem is that compared to other methods, ruled-based agents evaluation is not extendable. Manually building expert rules is difficult to implement in complex environments and may not meet diversity requirements.

**Random Agents.** Another choice is using random initial agents as evaluation partners (Strouse et al., 2021; Xue et al., 2022). The first problem is that the diversity of the random initialization cannot be ensure. Furthermore, evaluate agent diversity is not only presenting in low level behaviors but also need a high level performance (Cui et al., 2022). Random initialization lacks of a high level performance to ensure the evaluation pool is diverse enough.

