# OpenReview forum: "Quantifying Zero-shot Coordination Capability with Behavior Preferring Partners"
_ICLR.cc/2024/Conference — Submitted to ICLR 2024_

### Official Review · Reviewer_PVtL · 2023-10-19

**Soundness:** 2 fair
**Presentation:** 1 poor
**Contribution:** 2 fair
**Rating:** 3
**Confidence:** 4

**Summary:**

This paper aims to create a reliable and comprehensive evaluation method for zero-shot coordination. The authors design an evaluation workflow includes three stages. Firstly the method generates behavior preferring agents with corresponding BRs. Secondly representative policies are selected based on a BR-diversity. Finally the selected policies and their BRs are used to evaluate the ego policy. The overall framework has potential to provide a more effective metric (BR-Prox) and also indicates some shortcuts of overcooked scenarios.

**Strengths:**

1. The methodology of constructing a "diversity-complete" set of BRs is reasonable and meaningful.
2. The authors propose the BR-Prox metric to measure the performance of an ego policy under the ZSC manner and use this metric to benchmark previous methods on ZSC.
3. The authors propose an algorithm to construct an evaluation population by first generating adequate policies and then selecting diverse ones.

**Weaknesses:**

1. Though the authors claim that a "diversity-complete" set may be intractable for complex environments, the proposed methodology (generation and selection) fails to well adhere to the exact definition in the desiderata (Section 3.2) with the lack of revealing the gap with a real "diversity-complete" set.
2. The proposed metric and evaluation workflow lack necessary discussions with previous approaches (e.g., methods in Table 1).
3. The presentation of this paper is generally obscure. The authors involve a sort of techniques during the evaluation workflow but do not well explain the necessity and details (e.g., how to represent the behavior feature of a policy and reasons to involve event-based rewards).
4. The evaluated baselines seem not distinguishable in the evaluation setting while the authors do not provide further insights of what kinds of approaches are generally useful on ZSC.
5. The evaluation workflow is restricted to a two-agent form while previous ZSC methods can generalize to multi-agent settings.

**Questions:**

1. How can we extend the evaluation manner to multi-agent settings?
2. In Figure 2, how are the high-level behaviors visualized? What is the meaning of different data points in Figure 2?
3. In Figure3, why does the population diversity first rise and then drop with the population size increasing? As near 0 values mean linear correlation, why is the diversity low in a small population size?
4. Does the P-Div metric mean $\text{PD}$ on $\pi_i$ instead of $BR(\pi_i)$?
5. In Section 5.2, how do Figure 4 and 5 show "increasing population size contributes to the improvement of performance under the condition that the diversity of the population is also grown".
6. How do the event-based rewards contribute to behavior preferring policies? Specifically, how do the method design and adjust $w$? How is this approach related to proposed "skill-level diversity"?
7. At the end of Section 3.2, "by selecting earlier checkpoints of the evaluation partners, it is simple to acquire evaluation partners with diverse skill levels." What is the actual method of selecting earlier checkpoints of the evaluation partners?
8. How can we represent the behavior feature $\theta$ of a policy?

---

> ### Author Response · Authors · 2023-11-17
> **Response to Reviewer PVtL (1/5)**
>
> We appreciate the reviewer's insightful comments and suggestions, and we are glad to learn that you regard our method as reasonable and meaningful.
>
> ### 1. Relations between our method and "diversity-complete"
> > (Weakness 1) Though the authors claim that a "diversity-complete" set may be intractable for complex environments, the proposed methodology (generation and selection) fails to well adhere to the exact definition in the desiderata (Section 3.2) with the lack of revealing the gap with a real "diversity-complete" set.
>
> **Meeting the Desiderata**. We acknowledge that our evaluation workflow is approximating the 'diversity-complete' by meeting the desiderata as much as possible.
> - Reasonable: We restrict the reward space to indicate reasonable behaviors by adding the original task reward and filtering sabotaging agents after generating candidates.
> - Skill-style diverse: We traverse the reward space for diverse fully trained behavior-preferring agents and select the representative set by BR-Div.
> - Skill-level diverse: We collect the earlier checkpoints of the selected fully trained evaluation partners to increase the skill-level diversity.
>
> We highlight how our evaluation workflow meets the desiderata in Sec. 4.2 of the update paper.
>
> **Gap with "diversity-complete"**. First, we need to claim that we cannot get a real "diversity-complete" set of evaluation partners for most tasks. We reveal the gap of our generated partners with a real "diversity-complete" set as follows:
> - Events are designed based on human knowledge and may not cover all possible reasonable behaviors.
> - We can only obtain approximate BRs.
> - Style levels of different checkpoints may not cover all skill levels.
>
> Though our evaluation partners still have a gap with a real "diversity-complete", we are the first to propose the concept of ideal evaluation partners, i.e., "diversity-complete" and our evaluation workflow makes a significant improvement over previous evaluation methods in providing a fair and comprehensive evaluation.
>
>
> ### 2. Discussion between our method and previous methods
> > (Weakness 2) The proposed metric and evaluation workflow lack necessary discussions with previous approaches (e.g., methods in Table 1).
>
> We first clarify that we list the previous evaluation methods in the first column and the works that utilize them in the second column in Table 1. For the previous evaluation methods, we have already discussed their defects in Sec. 3.2 and Appendix C.
>
> Defects of previous evaluation partners:
> - Human Players: The human players in ZSC problem are 'perfect' candidates for evaluation because human are strictly qualified as unseen partners. While evaluating with human players are expensive, time-consuming and unrepeatable. We need a more efficient evaluation method as a supplement for human evaluation.
> - Human Proxy Agents: Human proxy agents in overcooked environment do not account for human behaviors [5], which shows that using human proxy agents does not represent the diversity of human.
> - Trained Self-play Agents: Train self-play agents are similar to the agents used to train the ego agent and are not diverse, as shown in Figure 2.
> - Rule-based Specialist: Manually building expert rules is difficult to implement in complex environments and may not meet diversity requirements. We point out that designing a set of events is much easier than designing a policy.
> - Random Agents: The diversity of the random initializations cannot be ensured and random initializations lack of high level performance.
>
> Defects of previous metric:
> - Cross-play with Trained Adapted Agents: Some works compare their ZSC methods by cross-playing the agents trained by other ZSC methods and their ego agents. This results include self-play performance and does not completely reflect the capabilities of ZSC. On the other hand, excluding the self-play results leads to potential unfairness.
>
> The major difference is that **previous evaluation partners and metrics are unable to provide a comprehensive and fair evaluation due to the evaluation partners are not diverse or the metric is unfair. Instead, our evaluation workflow provide a comprehensive and fair measurement of ZSC capability by assembling reasonable evaluation partners with diverse skill-styles and skill-levels and by proposing BR-Prox to assess the generalization capability and improvement potential.**
>
> We have highlighted the differences in Sec. 2 of the update paper. We also remark that we are the first to systematically investigate the construction of evaluation partners and the measurement of ZSC capability. Our aim is to ensure a more fair and comprehensive assessment of ZSC algorithms, addressing the current gaps in evaluation methodologies.

---

> > ### Author Response · Authors · 2023-11-17
> > **Response to Reviewer PVtL (2/5)**
> >
> > ### 3. Overcooked Evaluation Overhual
> > > (Weakness 4) The evaluated baselines seem not distinguishable in the evaluation setting while the authors do not provide further insights of what kinds of approaches are generally useful on ZSC.
> >
> > It is important to highlight that this 'undistinguishability' is, in fact, one of our significant findings. The observation that various baselines appear to be similar in performance within our evaluation framework is not a limitation but rather a crucial insight. This finding suggests that current evaluation layouts may not be sufficiently challenging or diverse to distinguish different ZSC methods. We argue that this discovery underscores the need for more complex and sophisticated evaluation environments and mechanisms in ZSC research.
> >
> > We categorized the current ZSC methods into two classes:
> > - co-play: a two-stage framework that pretrains a population via self-play and then trains the ego agent with the pre-trained population
> > - evolution: the ego agent is trained with an evolving population of partners.
> >
> > We point out that the most popular methods are mostly co-play methods, and under our evaluation workflow, co-play methods, especially MEP and HSP, are generally useful in ZSC. The conclusion is based on the results aggregated from all layouts and all population sizes, as shown in right of Figure 5 in the main text and Table 4 in the appendix.
> >
> > As a future work, we plan to introduce complex mechanisms into the Overcooked environment and apply our evaluation workflow to more environments to promote the development of ZSC methods.
> >
> > ### 4. Extend to Multi-agent setting
> > > (Weakness 5) The evaluation workflow is restricted to a two-agent form while previous ZSC methods can generalize to multi-agent settings.
> > (Question 1) How can we extend the evaluation manner to multi-agent settings?
> >
> >
> > To the best of our knowledge, current zero-shot coordination research (including FCP [1], TrajeDi [8], MEP [4], COLE [5], and HSP [2]) primarily involves theoretical frameworks and practical algorithms designed for two-player cooperative games. The most commonly used environments, such as Overcooked [6] and Hanabi [7], are also predominantly used in two-agent settings. While it is true that some ZSC research (including Other-Play [9], Any-Play [10]) can theoretically extend to n-player scenarios, the limitation of computational resources makes them evaluate the algorithms using 2-player settings in the experiments.
> >
> > Also, as discussed in previous ZSC research, the settings of more than two agents can be easily reduced to two-agent settings. It is not the key challenge of the ZSC problem. In the context of n agents collaborating, we can consider these n-1 partners to cooperate as a joint policy and the left agent as the other policy. Thus, we can train the behavior of preferring joint policies and their best responses in the same way, maintaining the effectiveness of our evaluation workflow.
> >
> > We strongly believe in the value of extending the evaluation environment to n-players, and we believe this will be the focus of our future work. We will consider implementing multi-agent setting environments and evaluations to contribute to promoting ZSC research in n-player settings. We are implementing our evaluation workflow in the 5-vs-5 Google Research Football [11], and we will update the additional results if time permits.

---

> ### Author Response · Authors · 2023-11-17
> **Response to Reviewer PVtL (3/5)**
>
> ### 5. Explanations
>
> Thank you for your suggestions on the improvement of our representation. We have added the explanations to the revised paper. We are glad to further discuss if you have other suggestions or questions about our paper.
>
> ### Explanations of the figures
>
> Due to the space limitation, details of our experiments can be found in Appendix B.
>
> #### a. Explanation of Fig. 2
> > (Question 2) In Figure 2, how are the high-level behaviors visualized? What is the meaning of different data points in Figure 2?
>
> To get Figure 2, we collect the occurrence of the designed events by counting the triggered events alongside the episodes and representing each trajectory as a vector containing the normalized times of event occurrences. Then we reduce the vectors into 2-dimension through principal components analysis (PCA [3]). Finally we plot the 2-dimension vectors in Figure 2. **Briefly, each data point means the normlized event occurrences of a trajectory.**
>
> #### b. Explanation of Fig. 3
> > (Question 3) In Figure 3, why does the population diversity first rise and then drop with the population size increasing? As near 0 values mean linear correlation, why is the diversity low in a small population size?
>
> Recall that we define the population diversity as $\operatorname{PD}(\\{\pi _{i}\\} _{i=1}^{M}) = \operatorname{det}(\bf{K})$, where $\bf{K} _{ij} = \theta _{i}\cdot\theta _{j}$ and each element in $\bf{K}$ is normalized into \[0,1\]. Thus, population diversity is a determinant. The determinant of a matrix represents the "volume" of the matrix but is also related to the dimension of the matrix.
>
> Therefore, the population diversity metric is related to both the diversity and the size of the population. Strictly speaking, we should only compare the points on the two curves with the same population size. The population diversity of a small population is low because it is the determinant of a small and normalized matrix.
>
> Figure 3 illustrates that sampling based on BR-Div reaches higher population diversity than that based on P-Div at the same population size before the population diversity diminishes, which means BR-Div is more effective in constructing a diverse population from a set of policies.
>
>
>
> #### c. Explanation of Fig. 4 and Fig. 5
> > (Question 5) In Section 5.2, how do Figure 4 and 5 show "increasing population size contributes to the improvement of performance under the condition that the diversity of the population is also grown".
>
> Thank you for your suggestion to improve our representation. We have updated the explanation in the revised paper.
>
> Specifically, in Figures 4 and 5, some algorithms get performance improvement with increasing population sizes in some layouts, which is exactly why we conclude that "increasing population size contributes to the improvement of performance under the condition that the diversity of the population is also grown". In the following, we explain what the condition is from two aspects.
> - Some ZSC methods lack an explicit mechanism to promote population diversity, including FCP and COLE. Thus, the performance of FCP and COLE does not benefit from increasing the population size from 24 to 36.
> - On the other hand, ZSC methods with explicit mechanisms to promote population diversity may sometimes fail to expand population diversity in some layouts. We provide a case study in the Counter Circ. and Forced Coord. layouts to show that the reason is existing ZSC methods may fail to effectively produce enough diverse, high-performing agents in some layouts.

---

> ### Author Response · Authors · 2023-11-17
> **Response to Reviewer PVtL (4/5)**
>
> ### Explanation of notions and concepts
> #### a. P-Div and BR-Div
> > (Question 4) Does the P-Div metric mean PD on $\pi _i$ instead of $BR(\pi _i)$?
>
> Yes, and P-Div is first mentioned in Sec. 4.1, "according to the partners population diversity (P-Div)". We further add the definition that $\operatorname{P-Div}(\\{\pi _{i}\\} _{i=1}^{M}) = \operatorname{PD}(\\{\pi _{i}\\} _{i=1}^{M})$ to make it clear.
>
> #### b. Behavior Feature Representation and Event-based Reward Design
> > (Weakness 3) The presentation of this paper is generally obscure. The authors involve a sort of techniques during the evaluation workflow but do not well explain the necessity and details (e.g., how to represent the behavior feature of a policy and reasons to involve event-based rewards).
>
> > (Question 6) How do the event-based rewards contribute to behavior preferring policies? Specifically, how do the method design and adjust $w$?
>
> > (Question 8) How can we represent the behavior feature $\theta$ of a policy?
>
> **Event-based Reward**. To clarify, we explain our motivation for introducing event-based rewards in Sec. 4.2. We expect partners to have diverse and reasonable behaviors, while the behavior space is intractably large. Inspired by [2] in which human preferences can be regarded as event-centric and modeled as event-based reward functions, we apply the event-based reward shaping method to encourage behavior discovery and generate behavior-preferring partners. The event-based reward space can be formulated as follows:
> $$
> \mathcal{R}=\\{r _{\bf{w}}| r _{\bf{w}}(s _{t}, \bf{a} _{t}) = r + \phi(s _{t}, \bf{a} _{t})^{T}\bf{w}, \bf{w} \in \mathbb{R}^{m}, \|\bf{w}\| _{\infty} \leq B _{\text{max}}, \sum _{i} \mathbb{1}(\bf w _{i} \neq 0) \leq C _{\text{max}}\\}
> $$
> The hyper-paramters $B _{\text{max}}$ and $C _{\text{max}}$ are used to restrict the reward space to indicate reasonable behaviors. $\phi(s _t,\bf{a} _{t})$ is the indicator of pre-defined events.
>
> **Policy Behavior Feature**. The policy behavior features that are used to figure out BR-Div depend on the environment and the task at hand, which are related to the skills being tested. For simplicity, we count the occurrence of the pre-defined events, i.e., $\mathbb{E}[\sum _{t=1}^{T}\phi(s _t,\bf{a} _t) ]$, as the policy behavior feature.
>
>
> **$\bf{w}$ Design**. Given the above formulation of reward space $\mathcal{R}$, we have restricted the space of $\bf{w}$ to indicate only reasonable combinations of events. $\mathcal{R}$ includes preferences that do not cause the game to be disrupted or unable to cooperate with. **Then we traverse the restricted reward space, i.e., the space of $\bf{w}$**. An event-based reward function $r _{w}$ indicates a behavior preference since $r _{w}$ encourages the preferred events (behaviors). The agent equipped with this event-based reward function is trained with an agent equipped with the reward according to the given task. The two-agent team approximates an NE in a two-player Markov game, which models the scenario in which a task-reward-maximizing agent collaborates with another agent or a human with its own preferences to complete the given task. Therefore, we get an agent preferring the behaviors presented by $r _{w}$ after approximating the NE.
>
>
>
> *An example of Overcooked*
>
> In Overcooked, we use $B _{\text{max}} = 20$, $C _{\text{max}}=3$ and generate up to 194 candidates and select up to 30 evaluation partners. The generated candidates are excluded if they cannot complete a delivery when cooperating with their BRs. The pre-defined events are listed as follows:
> | Events | Weights |
> | --- | --- |
> | Put an onion or a dish or a soup onto the counter  | 0  |
> | Pickup an onion or a dish or a soup from the counter | 0 |
> | Pickup an onion from the onion dispenser | -20,0,10  |
> | Pickup a dish from the dish dispenser  | -20,0,10 |
> | Pickup a soup | -20,0,5,10 |
> | Place an ingredient into the pot | -20,0,3,10 |
> | Deliver a soup | -20,0 |
> | Stay | -0.1,0,0.1 |
> | Movement | 0 |
> | Order Reward | 0.1,1 |
>
> We also provide the pseudocode for using these events to show that one can make a moderate effort to implement the event-based rewards.
> ```python=
> state = current state of the game
> action = action of the agent whose events are collected
> reward = 0
> for e_i in range(NUM_EVENTS):
>     if True == EVENT_JUDGER(e_i, state, action):
>         reward += EVENT_WEIGHTS[e_i]
> ```
> The details are also present in Appendix B.1.

---

> ### Author Response · Authors · 2023-11-17
> **Response to Reviewer PVtL (5/5)**
>
> **References**
>
> [1] DJ Strouse, Kevin McKee, Matt Botvinick, Edward Hughes, and Richard Everett. Collaborating with humans without human data. NeurIPS 2021.
>
> [2] Chao Yu, Jiaxuan Gao, Weilin Liu, Botian Xu, Hao Tang, Jiaqi Yang, Yu Wang, and Yi Wu. "Learning zero-shot cooperation with humans, assuming humans are biased". ICLR. 2023.
>
> [3] George H Dunteman. "Principal components analysis", volume 69. Sage, 1989.
>
> [4] Zhao, Rui, et al. "Maximum entropy population-based training for zero-shot human-ai coordination". AAAI. 2023.
>
> [5] Li, Yang, et al. "Cooperative Open-ended Learning Framework for Zero-shot Coordination." ICML. 2023.
>
> [6] Carroll, Micah, et al. "On the utility of learning about humans for human-ai coordination." NeurIPS. 2019.
>
> [7] Bard, Nolan, et al. "The hanabi challenge: A new frontier for ai research." Artificial Intelligence. 2020.
>
> [8] Lupu, Andrei, et al. "Trajectory diversity for zero-shot coordination." ICML, 2021.
>
> [9] Hu, Hengyuan, et al. "“other-play” for zero-shot coordination." ICML. 2020.
>
> [10] Lucas, Keane, and Ross E. Allen. "Any-Play: An Intrinsic Augmentation for Zero-Shot Coordination." AAMAS. 2022.
>
> [11] Kurach, Karol, et al. "Google research football: A novel reinforcement learning environment." AAAI. 2020.

---

> > ### Comment · Reviewer_PVtL · 2023-11-22
> > **Some followup comments**
> >
> > I thank the authors for make a thorough rebuttal that partly address my questions. Here are some followup comments about the authors' response that better express my concerns.
> >
> > ### Relations between the method and "diversity-complete"
> >
> > I acknowledge that implementing a real "diversity-complete" set can be hard. However, I cannot foresee a strong relation from the desiderata, e.g., the necessity of dividing diversity into skill-style diversity and skill-level diversity. Though the authors use BR-Div to select policies, it is doubtful that the construction process actually generate diverse population.
> >
> > Besides, can you show the process of that "we collect the earlier checkpoints of the selected fully trained evaluation partners to increase the skill-level diversity" in Algorithm 1?
> >
> > ### Discussion between our method and previous methods & Overcooked Evaluation Overhaul
> >
> > I think Table 1 is generally useful for clarifying related work. My concern is about how to compare different evaluation methods (e.g., policy construction methods) practically. Although you mention pros & cons of other methods (e.g., human players and self-play agents), I think a more significant work is to show what indeed Behavior Preferring Evaluation can bring (e.g., requirements of emerging behaviors that cannot be found by other evaluation methods). Evaluations and comparisons about this point can be helpful to illustrate the superiority of your method.
> >
> > ### Extend to Multi-agent Setting
> > I do not think splitting multiple agents into two groups is a trivial idea of MARL, but I believe that considering multi-agent case can be a future extension.
> >
> > ### About Behavior Feature Representation and Event-based Reward Design
> > The diversity of population generation seems highly depend on the design of events and weights. Probably the method may overfit the Overcooked environment. I used to think that the weights and behavior features are given from learning. The current approach, however, may restrict its use cases, as there can be various evaluation designs for an environment, with different implementations of events and weights.
> >
> > Besides, how do you choose the weights and what is the meaning of multiple weights for an event?

---

> > > ### Author Response · Authors · 2023-11-23
> > > **Response to Follow-up Comments (1/2)**
> > >
> > > ### Relations between the method and "diversity-complete"
> > >
> > > > I acknowledge that implementing a real "diversity-complete" set can be hard. However, I cannot foresee a strong relation from the desiderata, e.g., the necessity of dividing diversity into skill-style diversity and skill-level diversity.
> > >
> > > Ideal evaluation partners should be representative of the whole policy space. However, the whole policy space containing policies that have different skills is too large. Moreover, it is hard to directly increase the diversity of skills since skills can be different in many dimensions. Thus, we divide diversity into two dimensions to simplify this problem. Such a division has been implicitly used in recent ZSC works, including most of the works listed in Table 1.
> > >
> > > > Though the authors use BR-Div to select policies, it is doubtful that the construction process actually generate diverse population.
> > >
> > > We illustrate qualitative results to support the claim that "the construction process actually generates a diverse population" in Figures 2, 8, and 9.
> > >
> > > > Besides, can you show the process of that "we collect the earlier checkpoints of the selected fully trained evaluation partners to increase the skill-level diversity" in Algorithm 1?
> > >
> > > We have updated Algorithm 1 in the revised paper. We select the fully trained evaluation partners and then collect the earlier checkpoints to increase the skill-level diversity.
> > >
> > >
> > > ### Discussion between our method and previous methods & Overcooked Evaluation Overhaul
> > > > My concern is about how to compare different evaluation methods (e.g., policy construction methods) practically. ... . I think a more significant work is to show what indeed Behavior Preferring Evaluation can bring (e.g., requirements of emerging behaviors that cannot be found by other evaluation methods).
> > >
> > >
> > > Thanks for your suggestions. We provide comparisons of the high-level behaviors between self-play agents and our evaluation partners in Fig. 2, which shows that our method indeed generates significantly more diverse behaviors than self-play agents.
> > >
> > > And we also have visualizations of the high-level behaviors generated by our method in Figs. 8 and 9. As shown in the bottoms of Figs. 8 and 9, our method can learn some specific strategies that the other method cannot present, for example, in the Coord. Ring layout, while using the counter to deliver onions, dishes or soup cannot be learned in self-play agents, e.g., in populations constructed by MEP or trained apdative agents, e.g., COLE agents, our method can generate partners that use the counter more frequently.

---

> > > > ### Author Response · Authors · 2023-11-23
> > > > **Response to Follow-up Comments (2/2)**
> > > >
> > > > ### About Behavior Feature Representation and Event-based Reward Design
> > > > > Besides, how do you choose the weights and what is the meaning of multiple weights for an event?
> > > >
> > > > | Events | Weights |
> > > > | --- | --- |
> > > > | Put an onion or a dish or a soup onto the counter  | 0  |
> > > > | Pickup an onion or a dish or a soup from the counter | 0 |
> > > > | Pickup an onion from the onion dispenser | -20,0,10  |
> > > > | Pickup a dish from the dish dispenser  | -20,0,10 |
> > > > | Pickup a soup | -20,0,5,10 |
> > > > | Place an ingredient into the pot | -20,0,3,10 |
> > > > | Deliver a soup | -20,0 |
> > > > | Stay | -0.1,0,0.1 |
> > > > | Movement | 0 |
> > > > | Order Reward | 0.1,1 |
> > > >
> > > > Multiple weights for an event mean that we traverse the weights of the event to encourage different preferences for this event. For example, "-20,0" of the "Deliver a soup" event means generating agents that either refuse to deliver the soup to the serving point or have no preference for delivering the soup.
> > > >
> > > > We design the weights by trying some values to see whether the weights lead to expected behaviors in some layouts; thus, we further argue that "the construction process actually generates a diverse population".
> > > >
> > > > > The diversity of population generation seems highly depend on the design of events and weights. Probably the method may overfit the Overcooked environment. ... . The current approach, however, may restrict its use cases, as there can be various evaluation designs for an environment, with different implementations of events and weights.
> > > >
> > > > We clarify that the current event-based rewards and weights are the instantiation of the methodology in Overcooked. Moreover, event-based rewards and weights can be easily deployed in both complex robot tasks [1] and large multi-agent games [2, 3]. The event-based reward design is very similar to the feature and rewarding shaping, and thus one can make moderate effort to design the event-based reward and weights.
> > > >
> > > > We admit that the event-based rewards and weights have different implementations in one environment. Therefore, we design the event-based reward and weights to cover as much as possible reasonable behaviors to cover the different designs.
> > > >
> > > > To further address the overfitting concern of the event-based design and weights, we are incorporating the three Google Research Football scenarios "3 versus 1 with Keeper," "Pass and Shoot with Keeper," and "Run, Pass and Shoot with Keeper" into our evaluation workflow [2]. The events consist of short-passing, long-passing, high-passing, dribbling, and shooting the ball. We train agents with different preferences for attacking strategies as evaluation partners. We will update the additional results if time permits.
> > > >
> > > > > I used to think that the weights and behavior features are given from learning.
> > > >
> > > > Weights and event rewards can be obtained from learning methods. There indeed exist works that learn human behaviors without assuming a set of events [4] and we leave making these methods general to the ZSC domain as future work.
> > > >
> > > > We hope our explanation have addressed your concerns.
> > > >
> > > > **Reference**
> > > >
> > > > [1] Brohan, Anthony, et al. "Do as i can, not as i say: Grounding language in robotic affordances." CoRL. 2023.
> > > >
> > > > [2] Kurach, Karol, et al. "Google research football: A novel reinforcement learning environment." AAAI. 2020.
> > > >
> > > > [3] Deheng Ye, et al. Mastering Complex Control in MOBA Games with Deep Reinforcement Learning. AAAI. 2020.
> > > >
> > > > [4] Shah, Rohin, et al. "On the feasibility of learning, rather than assuming, human biases for reward inference." ICML. 2019.

---

> ### Author Response · Authors · 2023-11-23
> **Request for Consideration of Score Increase**
>
> Thank you for your valuable time and insightful feedback. We have made significant modifications to our paper based on your suggestions and thoroughly explained your concerns. Our work makes unique contributions to the zero-shot coordination problem, including the fact that we are the first to investigate the evaluation of ZSC capability and formally define the concepts of ideal evaluation partners. We propose an effective evaluation workflow and first provide a benchmark on the most used Overcooked environment. We believe these contributions are of significant relevance to research in this field.
>
> As you may be aware, unlike previous years, the discussion period this year can only last until November 22, and we are rapidly approaching this deadline.
>
> Considering the above improvements and contributions to our paper, we kindly request that you consider increasing the score of our manuscript. We believe these improvements reflect our commitment to quality and strongly enhance the overall quality of the paper.
>
> If you have any further questions or require more information to raise your score, please feel free to let us know.
>
> Thank you again for your time and consideration.
>
> Sincerely, authors

---

> > ### Comment · Reviewer_PVtL · 2023-12-05
> >
> > I thank the authors for providing detailed responses and careful revisions. Based on discussions above, I decide to keep my score. I do appreciate the authors' efforts in improving their work. However, I still believe this paper may need another review process. Currently my major concern is that this approach involves many sophisticated designs such as the event weights, making it more like an ad hoc method rather than an evaluation tool. I think that involving additional experiments on GRF, as the authors stated they will conduct later, and having a discussion on the influence of event & weight designs can really help evaluate the contributions.

---

### Official Review · Reviewer_tgST · 2023-10-29

**Soundness:** 2 fair
**Presentation:** 3 good
**Contribution:** 3 good
**Rating:** 6
**Confidence:** 5

**Summary:**

The paper proposes a new evaluation metric for the zero-shot coordination problem. The method involves first training diverse policies and the corresponding best responses, employing the similar method as HSP, and then select evaluation partners based on the best response diversity metric.
The ego agent is then evaluated with the selected partners and a metric called best response proximity is calculated based on the performance of the ego agent versus the best response policy. The experiment results demonstrate that some widely used layouts in the literature may lack enough complexity to evaluate the effectiveness of different methods.

**Strengths:**

1. In general, the authors propose an important question that the evaluation protocol should be improved in the literature of ZSC problem.
2. The idea that uses a set of sufficiently diverse policies as evaluation partners is straightforward and promising.
3. The experiment result demonstrates the effectiveness of the proposed metric to distinguish different methods in conflicts layouts.

**Weaknesses:**

A significant limitation is the absence of a crucial baseline, specifically HSP. Since the paper's approach to training evaluation candidates and best response policies closely mirrors that of HSP, it should not pose a substantial challenge to also train an ego agent for HSP. I would consider increasing my score if this limitation is addressed.

**Questions:**

1. In figure 3, how would the population diversity of selection with population diversity lower than that of selection with BR-Div? This seems to be counterintuitive as it is expected to find the subset with largest population diversity if P-Div is used as the selection metric.
2. Please explain the difference among different layouts in more details.  Are there other kinds of events except conflicts should be considerred to influence the perfomance of different methods in the introduced layouts?

---

> ### Author Response · Authors · 2023-11-17
> **Response to Reviewer tgST (1/2)**
>
> We appreciate the reviewer's insightful comments and suggestions, and we are glad to learn that you regard the evaluation of ZSC capability as an important problem.
>
> ### 1. Comparing HSP
> > (Weakness) A significant limitation is the absence of a crucial baseline, specifically HSP. Since the paper's approach to training evaluation candidates and best response policies closely mirrors that of HSP, it should not pose a substantial challenge to also train an ego agent for HSP. I would consider increasing my score if this limitation is addressed.
>
> Thank you for suggesting that HSP be included in the Overcooked evaluation. The HSP algorithm has been implemented and included in the comparison. We have updated the HSP results in figures 4, 5, 10, and 11. We present the performance rank under our evaluation workflow as follows for your convenience. Table 4 of Appendix B.3 contains the rankings as well.
>
> | Method \ Rank | 1 | 2 | 3 | 4 | 5 | 6 |
> | --- | --- | --- | --- | --- | --- | --- |
> | SP     | 0.0 | 5.56 | 11.11 | 5.56 | 5.56 | 72.22 |
> | FCP  |  5.56 | 16.67 | 44.44 | 5.56 | 22.22 | 5.56  |
> | MEP  |  38.89 | 22.22 | 22.22 | 5.56 | 11.11 | 0.0  |
> | TrajDi | 11.11 | 16.67 | 16.67 | 33.33 | 5.56 | 16.67 |
> | COLE   | 0.0 | 11.11 | 0.0 | 27.78 | 55.56 | 5.56   |
> | HSP | 44.44 | 27.78 | 5.56 | 22.22 | 0.0 | 0.0|
>
> The table shows the percentage of ranks that ZSC methods achieve; for example, HSP ranks first at 44.44 percent of all layouts and population sizes. The results show that HSP outperforms the other algorithms evaluated.
>
> ### 2. BR-Div is counterintuitive?
> > (Question 1) In figure 3, how would the population diversity of selection with population diversity lower than that of selection with BR-Div? This seems to be counterintuitive as it is expected to find the subset with largest population diversity if P-Div is used as the selection metric.
>
> BR-Div is a novel diversity metric for population diversity in the ZSC domain, as we demonstrate. We argue that the diversity of a set of cooperative agents' best responses should be used to measure population diversity.
>
> The insight is that the ego agent strives to emulate the BRs to any partner in the training population. As a result, **an ego agent with higher ZSC capability emulates more BRs**. To evaluate the ZSC capability, we should measure the ego agent's ability to emulate a large number of BRs, i.e., expose the ego agent to partners with a large number of BRs.
>
> We verify the effectiveness of BR-Div in Figure 2, Figure 3, Figure 8 and Figure 9, both qualitatively and quantitatively. **A simple and intuitive explanation of why BR-Div is more effective than P-Div is that two different partners may respond to similar BRs or even the same BR.** This phenomenon has also been observed in recent works [1,2].

---

> > ### Author Response · Authors · 2023-11-17
> > **Response to Reviewer tgST (2/2)**
> >
> > ### 3. The difference among layouts in more details
> > > (Question 2) Please explain the difference among different layouts in more details. Are there other kinds of events except conflicts should be considerred to influence the perfomance of different methods in the introduced layouts?
> >
> > When we talk about the baseline's performance, we need to note that we first found the issue of not enough discrimination in the layouts of the original overcooked environment. Our evaluation workflow reflects the defects of previous evaluation environments' layouts. Appendix A contains a detailed description of the layouts. We will introduce them briefly below.
> > - Forced Coordination: Because the two players are in separate, non-overlapping sections of the kitchen, they must cooperate.
> > - Counter Circuit: Players must coordinate their movements and make use of the middle counter.
> > - Coordination Ring: Players must coordinate their movements in this ring.
> > - Bothway Coordination: allows both left and right agents access to onions and pots, providing them with more policy space and cooperation styles.
> > - Blocked Corridor: A U-shaped corridor forces the agents to work together to reduce the time it takes to prepare the soup. This layout includes a variety of coordination styles, such as using spare counters and coordinating agent movement.
> > - Asymmetric Advantages and Asymmetric Coordination: Two agents have individual advantages in completing some tasks, and they must coordinate their advantages in order to maximize their performance.
> >
> > We know that other things, like the layout size, the number of possible coordination conventions, and the skill levels of the agents, may have an effect on how well baselines do (see Sec. 5.1 for differences in performance with evaluation partners of different skill levels). However, we discovered that **the resource and physical collisions between players are the most influential factors in evaluating ZSC capability, which are also two of the main reasons for the other factors mentioned above**. When conflicts exist in layouts, the need for coordination capabilities is much greater than when they do not. This is due to the fact that the strategy is more influenced by other players at this time, such as the preferences of other players' strategies and the coordination capabilities of other players who are in conflict.
> >
> > We hope that the added results can address the limitations of our paper and that our explanation can address your concerns. We are looking forward to your reply.
> >
> > **Reference**
> >
> > [1] Sarkar, Bidipta, Andy Shih, and Dorsa Sadigh. "Diverse Conventions for Human-AI Collaboration." NeurIPS. 2023.
> >
> > [2] Rahman, Arrasy, Jiaxun Cui, and Peter Stone. "Minimum Coverage Sets for Training Robust Ad Hoc Teamwork Agents." arXiv preprint arXiv:2308.09595 (2023).

---

> ### Comment · Reviewer_tgST · 2023-11-21
>
> As the authors present additional experiment results to resolve my reservations, especially incorporating HSP as a baseline, I decided to raise my score.

---

### Official Review · Reviewer_dEYW · 2023-11-01

**Soundness:** 2 fair
**Presentation:** 3 good
**Contribution:** 2 fair
**Rating:** 5
**Confidence:** 4

**Summary:**

This paper studies the evaluation metric for zero-shot coordination. The authors propose to construct diverse evaluation partners with their approximate BRs, and then compute the proposed BR-Prox across these evaluation partners as the metric. BR-Prox measures the performance similarity between the ego agent and the approximate BRs of the evaluation partners.

**Strengths:**

- The studied question is important and interesting for zero-shot coordination
- The paper is easy to follow

**Weaknesses:**

- The main concern is the practicality of the proposed evaluation method. It evolves complicated steps to construct such evaluation agents and prepare their approximate BRs. The idea of computing the BR diversity is straightforward, but it is not easy to really use such a metric in practice. And the huge implementation effort would definitely reduce its impact on the community.
- The design of reward space is crucial for the proposed evaluation partners. However, it is clearly discussed, at least I didn’t find it yet in the main text. What’s more, it is hard to say the resulting partners would show expected reasonable behaviors.
- In addition to the evaluation agents, the proposed method requires users to compute the approximate BRs. In simple tasks like Overcooked, it could be fine. However, it could be hard to obtain in complex robot tasks.

**Questions:**

- How to compute the approximate BR, $\widehat{BR}(\pi_{\omega})$ ?
- How to design the reward space?
- If all of these evaluation agents can constructed, why not use them to train the ego agents?

---

> ### Author Response · Authors · 2023-11-17
> **Response to Reviewer dEYW (1/3)**
>
> We appreciate the reviewer's insightful comments and are glad to learn that you regard our work as important and interesting. We understand the concerns about practicality when applying our evaluation workflow to complex tasks. We want to clarify that the additional implementation effort is not as heavy as intuitively understood, and we have opened our code and models to reduce the implementation effort.
>
> **We first explain the practicality of computing approximate BRs and describe how we design the reward space, then further address the concern about practicality.**
>
> ### 1. Practicality of Approximate BR
> > (Question 1) How to compute the approximate BR?
>
> > (Weakness 3) In addition to the evaluation agents, the proposed method requires users to compute the approximate BRs. In simple tasks like Overcooked, it could be fine. However, it could be hard to obtain in complex robot tasks.
>
> **Computing approximate BR.** We have detailed discussed why training approximate BRs is practical for complex tasks in the common response. Our evaluation workflow includes two kinds of approximate best responses. We further explain how we compute the approximate BR here.
>
> The first ones are obtained when approximating the Nash Equilibrium (NE) of the two-player Markov game constructed with event-based rewards. As described in the paragraph Event-based Behavior Preferring Partners of Sec. 4.2, the two agents receiving an event-based reward and the original task reward, respectively, construct a two-player Markov game. Then, each agent works on their own to maximize their own rewards using the Proximal Policy Optimization (PPO) algorithm. This converges to the NE of this two-player Markov game because the event-based reward still encourages the agents to work together [1]. Then we obtain an approximate NE, which consists of two policies that approximate the best responses to each other. Thus, we obtain a fully trained behavior-preferring partner and its approximate BR. **It won't be more difficult than training the ZSC methods in this case because approximating the NE is like training a self-play agent-pair**.
>
> The second ones are the approximate best responses to earlier checkpoints of the fully trained behavior preferring agents. We train an approximate BR by performing the PPO algorithm to maximize the original task reward with the partner as one of those earlier checkpoints. **The complexity of this case will be lower than training a self-play agent pair since one of the agents is fixed.**
>
> Therefore, we would like to alleviate the concern raised about the feasibility of obtaining Approx. BRs in more complex tasks, such as robotics. We argue that training partner agents and Approx. BRs could be more practical than intuitively thought and more convenient and practical compared to frequently recruiting humans for evaluations.
>
> If an algorithm is capable of training two robots to collaboratively solve complex tasks, i.e., at least capable of training a self-play agent pair to solve complex tasks, **the extension of this capability to approximate an NE or train strong responses to fixed partners should be viewed as practically achievable**. The creation of these kinds of algorithms naturally shows that they can handle the computational and training problems that come with making approximate BRs.
>
> Moreover, it's important to understand that we only approximate the BRs instead of finding the exact BRs in our framework. The approximate BRs serve primarily as strong responses for each evaluation partner. The implementation and computation requirements for approximating BRs are much easier than finding exact BRs. With feasible implementation and computation costs, approximate BRs benefit the ZSC evaluation by supporting the BR-Div and providing a fair and robust measurement for ZSC performance.

---

> ### Author Response · Authors · 2023-11-17
> **Response to Reviewer dEYW (2/3)**
>
> ### 2. Reward Space Design
> > (Question 2) How to design the reward space?
>
> > (Weakness 2) The design of reward space is crucial for the proposed evaluation partners. However, it is clearly discussed, at least I didn’t find it yet in the main text. What’s more, it is hard to say the resulting partners would show expected reasonable behaviors.
>
> We have presented our reward-space design in Sec. 4.2, and we further give the details of reward-space in Overcooked in Sec. 5. Specifically, the event-based reward space can be formulated as follows:
> $$
> \mathcal{R}=\\{r _{\bf{w}}| r _{\bf{w}}(s _{t}, \bf{a} _{t}) = r + \phi(s _{t}, \bf{a} _{t})^{T}\bf{w}, \bf{w} \in \mathbb{R}^{m}, \|\bf{w}\| _{\infty} \leq B _{\text{max}}, \sum _{i} \mathbb{1}(\bf w _{i} \neq 0) \leq C _{\text{max}}\\}
> $$
> The hyper-paramters $B _{\text{max}}$ and $C _{\text{max}}$ are used to restrict the reward space to indicate reasonable behaviors. $\phi(s _t,\bf{a} _{t})$ is the indicator of pre-defined events.
>
> Event-based rewards have been widely used in lots of complex tasks, including robotics [2], large-scale games [3,4] and product management [5]. **We acknowledge that event-based rewards cannot promise to generate all the expected behaviors, while we can indeed get diverse reasonable behaviors after traversing the designed reward space in Overcooked.**
>
> **We provide visualizations of learned behaviors in Figures 8 and 9 to show that we indeed learned diverse, reasonable behaviors.**
>
> Moreover, we give an example of how to design the reward space.
>
> ### An example of Overcooked
> In Overcooked, we use $B_{\text{max}} = 20$, $C_{\text{max}}=3$ and generate up to 194 candidates and select up to 30 evaluation partners. The generated candidates are excluded if they cannot complete a delivery when cooperating with their BRs. The pre-defined events are listed as follows:
> | Events | Weights |
> | --- | --- |
> | Put an onion or a dish or a soup onto the counter  | 0  |
> | Pickup an onion or a dish or a soup from the counter | 0 |
> | Pickup an onion from the onion dispenser | -20,0,10  |
> | Pickup a dish from the dish dispenser  | -20,0,10 |
> | Pickup a soup | -20,0,5,10 |
> | Place an ingredient into the pot | -20,0,3,10 |
> | Deliver a soup | -20,0 |
> | Stay | -0.1,0,0.1 |
> | Movement | 0 |
> | Order Reward | 0.1,1 |
>
> We also provide the pseudocode for using these events to show that one can make a moderate effort to implement the event-based rewards.
> ```python=
> state = current state of the game
> action = action of the agent whose events are collected
> reward = 0
> for e_i in range(NUM_EVENTS):
>     if True == EVENT_JUDGER(e_i, state, action):
>         reward += EVENT_WEIGHTS[e_i]
> ```
> The details are also present in Appendix B.1.
>
> ### 3. Practicality Concerns
> > (Weakness 1) The main concern is the practicality of the proposed evaluation method. It evolves complicated steps to construct such evaluation agents and prepare their approximate BRs. ... . And the huge implementation effort would definitely reduce its impact on the community.
>
> **We have clarified that the steps, including designing event-based rewards, generating evaluation partners, and training approximate BRs, are practical in the above replies.**
>
> We point out that in the rising ZSC community, Overcooked is the most popular and most used for algorithm evaluation, while the community lacks a fair and comprehensive evaluation method. Our evaluation workflow is a strategic enhancement that promises to deliver a fair and comprehensive evaluation of the ZSC algorithm for the current tasks at hand. **Understanding the potential complexity of this implementation, we open-source our code and train agent policies to make a conscious effort to alleviate the burden on the following researchers.**
>
> It's crucial to emphasize that our method is far from impractical in complex environments like robotics. **In scenarios where the algorithm under evaluation successfully trains an agent capable of completing the task, incorporating our evaluation process is entirely feasible and beneficial.** The idea of an event-based reward adds the chance for human involvement within the set parameters of evaluation, giving you more precise control over the situations in which the algorithm is evaluated. For instance, if the objective is to enhance an agent's adaptability to various human reaction speeds, we can effectively design event-based rewards to create evaluation partners with differing response times. Moreover, building a reliable evaluation method is a prerequisite for the vigorous development of the ZSC community. As a result, even though we acknowledge that any new evaluation method will inevitably require some implementation work, we think that the potential advantages of using our framework outweigh the practical difficulties. By providing our implementation and offering ready-to-use components, we aim to enhance the accessibility and impact of our evaluation method within the ZSC research community.

---

> ### Author Response · Authors · 2023-11-17
> **Response to Reviewer dEYW (3/3)**
>
> ### 4. Novelty of BR Diversity
> > (Weakness 1) The idea of computing the BR diversity is straightforward.
>
> We emphasize that we propose a novel metric for population diversity in the ZSC domain, BR-Div. We argue that the measurement of population diversity among a group of cooperative agents should be based on the diversity exhibited in their best responses. The idea is that the ego agent strives to replicate the behavior of the best response (BR) while interacting with any partner within the training population.
>
> **Thus an ego agent with better ZSC capability means that the ego agent emulates more BRs. To evaluate the ZSC capability, we should measure that capability of emulating a lot of BRs**, i.e., expose the ego agent to partners with a lot of BRs.
>
> We verify the effectiveness of BR-Div via experiments, and the results are shown in Figure 2, Figure 3, Figure 8, and Figure 9, both qualitatively and quantitatively. A simple and intuitive explanation of why BR-Div is more effective than the population diversity of partners is that two different partners may correspond to similar BRs or even the same BR [6].
>
>
> ### 5. Why not use the evaluation partners to train the ego agents?
> > (Question 3) If all of these evaluation agents can constructed, why not use them to train the ego agents?
>
> To clarify, our primary focus is on developing an evaluation methodology with a keen emphasis on the fairness of comparisons. **The key challenge of the ZSC problem is generating ego agents that can cooperate with unknown partners.** Thus, we generate a set of diverse evaluation partners, evaluate the agents with these evaluation partners, and measure the ZSC performance with BR-Prox. The problem of how to train ZSC agents is beyond our scope.
>
> We discuss this problem in two cases:
> - Under our evaluation workflow, training an ego agent using the evaluation partners and evaluating the trained ego agent with the same set of evaluation partners is similar to using the test dataset as the training data in the supervised learning algorithms, which is unreasonable.
> - However, if you mean training the ego agent with a set of diverse partners constructed using our method and evaluating the trained ego agent with another set of constructed agents, we have added similar experiments to the updated paper. HSP is an algorithm that constructs partners to simulate human behaviors and trains the ego agent with those constructed partners. We train the HSP ego agents with unselected evaluation partner candidates and evaluate the HSP ego agents with the selected evaluation partners.
>
> We hope our reply addresses your concerns about practicality, and we are glad to discuss further if you have any other suggestions or questions.
>
> **Reference**
>
> [1] Ding, Dongsheng, et al. "Independent policy gradient for large-scale markov potential games: Sharper rates, function approximation, and game-agnostic convergence." ICML. 2022.
>
> [2] Brohan, Anthony, et al. "Do as i can, not as i say: Grounding language in robotic affordances." CoRL. 2023.
>
> [3] Kurach, Karol, et al. "Google research football: A novel reinforcement learning environment." AAAI. 2020.
>
> [4] Deheng Ye, et al. Mastering Complex Control in MOBA Games with Deep Reinforcement Learning. AAAI. 2020.
>
> [5] Shi, Zhenyu, et al. "Learning expensive coordination: An event-based deep RL approach." ICLR. 2020.
>
> [6] Sarkar, Bidipta, Andy Shih, and Dorsa Sadigh. "Diverse Conventions for Human-AI Collaboration." NeurIPS. 2023.

---

> > ### Comment · Reviewer_dEYW · 2023-11-22
> > **Thanks for your effort**
> >
> > I appreciate the authors' effort in the discussion, especially regarding the complexity of the proposed method. I think I understand what you mean by saying "The complexity of this case will be lower than training a self-play agent pair". However, based on my understanding of the whole pipeline of the proposed method, I don't believe the method is easy to use or transfer to other domains except the overcooked. Anyway, I thank your effort and would raise my score to weak rejection.

---

> > > ### Author Response · Authors · 2023-11-22
> > > **Thanks and Further Effort**
> > >
> > > Thank you for raising the score. We specify that our method is implemented in Overcooked, as the majority of ZSC methods are created and evaluated in Overcooked.
> > >
> > > We are incorporating the three Google Research Football scenarios "3 versus 1 with Keeper," "Pass and Shoot with Keeper," and "Run, Pass and Shoot with Keeper" into our evaluation workflow [1]. The events consist of short-passing, long-passing, high-passing, dribbling and shotting the ball. We train agents with different preferences of attacking strategies as evaluation partners. We will update the additional results if time permits.
> > >
> > > [1] Kurach, Karol, et al. "Google research football: A novel reinforcement learning environment." AAAI. 2020.

---

### Official Review · Reviewer_6Vtg · 2023-11-13

**Soundness:** 3 good
**Presentation:** 3 good
**Contribution:** 3 good
**Rating:** 6
**Confidence:** 2

**Summary:**

The performance of the agent's Zero Shot Coordination (ZSC) capability is difficult to measure and quantify. The difficulties are twofold: (1) how to construct sufficient diverse evaluation partners? (2) how to measure the performance? Most previous methods focus on designing superior ZSC algorithms while not paying much attention to the evaluation metric.

This paper first proposes to construct 'diversity-complete' evaluation partners by maximizing the best response diversity (the population diversity of the BRs to the evaluation partners). Then the paper proposes a Best Response Proximity (BR-Prox) metric, which quantifies the ZSC capability as the performance similarity to each evaluation partner’s approximate best response, demonstrating generalization capability and improvement potential.

Evaluations conducted on the overcooked environment validate the effectiveness of the proposed evaluation workflow and show some interesting results.

**Strengths:**

* The paper may be the first to systematically study how to measure and quantify the agent's Zero Shot Coordination (ZSC) capability. The proposed evaluation workflow is technically sound.
* Sufficient experiments are designed to demonstrate the effectiveness of the method. The results find that the most used layouts in the overcooked environment cannot show the ZSC capability difference among the ZSC methods.
* Overall, the writing of the article is relatively clear.

**Weaknesses:**

* Some parts of the paper are not very clear. For example, the motivation for introducing the event-based rewards is not clear.
* In practical implementation, the method requires humans to manually define some triggered events so as to derive diverse behaviors, which is difficult to obtain in complex tasks.
* Comparisons with previous baselines listed in Table 1 may be missing.

**Questions:**

Please see the weakness above.

---

> ### Author Response · Authors · 2023-11-17
> **Response to Reviewer 6Vtg (1/2)**
>
> We appreciate the reviewer's insightful comments and suggestions, and we are glad to learn that you consider our work sound and interesting.
>
> ### 1. Event-based reward
> > （Weakness 1）Some parts of the paper are not very clear. For example, the motivation for introducing the event-based rewards is not clear.
>
> Thank you for suggesting potential improvements to the presentation. We have elaborated on some technical details in the updated paper.
>
> Specifically, we explain our motivation for introducing event-based rewards in Sec. 4.2. We expect partners to have diverse and reasonable behaviors, while the behavior space is intractably large. Inspired by [1] in which human preferences can be regarded as event-centric and modeled as event-based reward functions, we apply the event-based reward shaping method to encourage behavior discovery and generate behavior preferring partners.
>
> > (Weakness 2) In practical implementation, the method requires humans to manually define some triggered events so as to derive diverse behaviors, which is difficult to obtain in complex tasks.
>
> We have detailedly discussed the practicality of our evaluation method in the common response. Specifically, we argue that the extra human effort is not as heavy as intuitively thought when applying our evaluation workflow to a new task. Defining some triggered events in complex tasks is very similar to conventional feature and reward engineering, which is inevitable when exploring new tasks. Moreover, event-based reward shaping has already been applied in both complex robot tasks [2] and large multi-agent games [3,4]. So defining events may not cost a lot of extra effort and is general to the ZSC domain. Besides, there indeed exist works that learn human behaviors without assuming a set of events [5] and we leave making these methods general to the ZSC domain as future work.
>
> We additionally give an example of the event-based reward design in Overcooked to explain the feasibility of designing event-based rewards. The pre-defined events are listed as follows:
>
> | Events | Weights |
> | --- | --- |
> | Put an onion or a dish or a soup onto the counter  | 0  |
> | Pickup an onion or a dish or a soup from the counter | 0 |
> | Pickup an onion from the onion dispenser | -20,0,10  |
> | Pickup a dish from the dish dispenser  | -20,0,10 |
> | Pickup a soup | -20,0,5,10 |
> | Place an ingredient into the pot | -20,0,3,10 |
> | Deliver a soup | -20,0 |
> | Stay | -0.1,0,0.1 |
> | Movement | 0 |
> | Order Reward | 0.1,1 |
>
> We also provide the pseudocode for using these events to show that one can make a moderate effort to implement the event-based rewards.
> ```python=
> state = current state of the game
> action = action of the agent whose events are collected
> reward = 0
> for e_i in range(NUM_EVENTS):
>     if True == EVENT_JUDGER(e_i, state, action):
>         reward += EVENT_WEIGHTS[e_i]
> ```
> The details are also present in Appendix B.1.

---

> ### Author Response · Authors · 2023-11-17
> **Response to Reviewer 6Vtg (2/2)**
>
> ### 2. Comparison with previous evaluation methods
> > (Weakness 3) Comparisons with previous baselines listed in Table 1 may be missing.
>
> We do not fully understand whether you mean comparison between our evaluation workflow and the previous evaluation methods or comparison among the ZSC algorithms.
>
> **Comparisons with Previous Evaluation Methods**
> We detailedly compare the previous evaluation methods and our evaluation workflow in the common response, which reveals that previous evaluation partners and metrics cannot provide a comprehensive and fair evaluation and that our evaluation workflow has advantages in measuring the ZSC capability.
>
> We first clarify that we list the previous evaluation methods in the first column and the works that utilize them in the second column in Table 1.
>
> **For the previous evaluation methods**, we have already discussed their defects in Sec. 3.2 and Appendix C.
>
> Defects of previous evaluation partners:
> - Human Players: The human players in ZSC problem are 'perfect' candidates for evaluation because human are strictly qualified as unseen partners. While evaluating with human players are expensive, time-consuming and unrepeatable. We need a more efficient evaluation method as a supplement for human evaluation.
> - Human Proxy Agents: Human proxy agents in overcooked environment do not account for human behaviors [5], which shows that using human proxy agents does not represent the diversity of human.
> - Trained Self-play Agents: Train self-play agents are similar to the agents used to train the ego agent and are not diverse, as shown in Figure 2.
> - Rule-based Specialist: Manually building expert rules is difficult to implement in complex environments and may not meet diversity requirements. We point out that designing a set of events is much easier than designing a policy.
> - Random Agents: The diversity of the random initializations cannot be ensured and random initializations lack of high level performance.
>
> Defects of previous metric:
> - Cross-play with Trained Adapted Agents: Some works compare their ZSC methods by cross-playing the agents trained by other ZSC methods and their ego agents. This results include self-play performance and does not completely reflect the capabilities of ZSC. On the other hand, excluding the self-play results leads to potential unfairness.
>
> We have highlighted the differences in Sec. 2 of the update paper. We also remark that we are the first to systematically investigate the construction of evaluation partners and the measurement of ZSC capability.
>
> **Comparison among ZSC algorithms**
> To improve the reliability of our evaluation results, we have reevaluated five representative ZSC algorithms: SP, FCP, MEP, TrajeDi, and COLE. We also reevaluate the HSP [1] algorithm in Overcooked and add the results to the revised paper.
>
> We hope that our explanations and clarifications have addressed your concerns and shed light on the insights behind our methodology and its potential impact. We invite further discussion and are open to any additional questions or suggestions you may have.
>
> **Reference**
>
> [1] Chao Yu, Jiaxuan Gao, Weilin Liu, Botian Xu, Hao Tang, Jiaqi Yang, Yu Wang, and Yi Wu. "Learning zero-shot cooperation with humans, assuming humans are biased". ICLR. 2023.
>
> [2] Brohan, Anthony, et al. "Do as i can, not as i say: Grounding language in robotic affordances." CoRL. 2023.
>
> [3] Kurach, Karol, et al. "Google research football: A novel reinforcement learning environment." AAAI. 2020.
>
> [4] Deheng Ye, et al. Mastering Complex Control in MOBA Games with Deep Reinforcement Learning. AAAI. 2020.
>
> [5] Shah, Rohin, et al. "On the feasibility of learning, rather than assuming, human biases for reward inference." ICML. 2019.

---

> ### Author Response · Authors · 2023-11-23
> **A Gentle Reminder for Rebuttal Consideration**
>
> Thanks again for your valuable comments and suggestions. We have submitted our rebuttals to your reviews. We explain the problems and try our best to settle your concerns, including paper revisions.
>
> Our work makes unique contributions to the zero-shot coordination problem, including the fact that we are the first to investigate the evaluation of ZSC capability and formally define the concepts of ideal evaluation partners. We propose an effective evaluation workflow and first provide a benchmark on the most used Overcooked environment. We believe these contributions are of significant relevance to research in this field.
>
> As you may be aware, unlike previous years, the discussion period this year can only last until November 22, and we are rapidly approaching this deadline.
>
> Considering the above improvements and contributions to our paper, we kindly request that you consider increasing the score of our manuscript if our revised manuscript and rebuttal more closely meet your expectations for the paper.
>
> If you have any further questions or require more information to raise your score, please feel free to let us know. We look forward to your response and are eager to continue our discussion.
>
> Sincerely, Authors

---

> > ### Comment · Reviewer_6Vtg · 2023-12-05
> > **Official Comment by Reviewer 6Vtg**
> >
> > I am very grateful for the author's meticulous and detailed responses. The author's replies have addressed most of my concerns. However, I am not particularly familiar with Zero-shot Coordination. Considering the concerns raised by other reviewers regarding the applicability of the method in more complex scenarios, I keep my score currently.

---

### Author Response · Authors · 2023-11-17
**Common Response (1/3)**

We gratefully acknowledge all reviewers for their insightful comments and suggestions. Three critical issues summarised from the comments and suggestions in the common response will be discussed as follows. The remaining issues will be addressed separately.

## Motivation of Our Evaluation Workflow
The task of developing agents capable of achieving Zero-Shot Coordination (ZSC) with unfamiliar partners poses a novel challenge within the field of cooperative Multi-Agent Reinforcement Learning (MARL). To clarify, the development of an evaluation methodology for ZSC capability is our primary objective, with a particular emphasis on ensuring comprehensiveness and fariness in comparisons. To clarify, the development of an evaluation methodology for ZSC capability is our primary objective, with a particular emphasis on ensuring comprehensiveness and impartiality in comparisons. Following a discussion of the reasons why current evaluation methods may not yield a comprehensive and fair comparison, we select the most direct and fair approach to assess the capability of ZSC: each trained ego agent is evaluated using a consistent set of fixed evaluation companions. Following this, we quantify the ZSC capability as BR-Prox and address the issue of measuring it comprehensively by approximating a "diversity-complete" set of evaluation partners.

### Problems in Previous Evaluation Methods
Reviewers (6Vtg and PVtL) have mentioned that we miss the discussion about the previous evaluation methods. We first clarify that we list the previous evaluation methods in the first column and the works utilize them in the second column in Table 1. The shortcomings of the preceding evaluation methodologies have been previously examined in Section 3.2 and Appendix C.

Defects of previous evaluation partners:
- Human Players: The human players in ZSC problem are 'perfect' candidates for evaluation because human are strictly qualified as unseen partners. While evaluating with human players are expensive, time-consuming and unrepeatable. We need a more efficient evaluation method as a supplement for human evaluation.
- Human Proxy Agents: Human proxy agents in overcooked environment do not account for human behaviors [5], which shows that using human proxy agents does not represent the diversity of human.
- Trained Self-play Agents: Train self-play agents are similar to the agents used to train the ego agent and are not diverse, as shown in Figure 2.
- Rule-based Specialist: Manually building expert rules is difficult to implement in complex environments and may not meet diversity requirements. We point out that designing a set of events is much easier than designing a policy.
- Random Agents: The diversity of the random initializations cannot be ensured and random initializations lack of high level performance.

Defects of previous metric:
- Cross-play with Trained Adapted Agents: Some works compare their ZSC methods by cross-playing the agents trained by other ZSC methods and their ego agents. This results include self-play performance and does not completely reflect the capabilities of ZSC. On the other hand, excluding the self-play results leads to potential unfairness.

The major difference is that **previous evaluation partners and metrics are unable to provide a comprehensive and fair evaluation due to the evaluation partners are not diverse or the metric is unfair. Instead, our evaluation workflow provide a comprehensive and fair measurement of ZSC capability by assembling reasonable evaluation partners with diverse skill-styles and skill-levels and by proposing BR-Prox to assess the generalization capability and improvement potential.**

The distinctions are highlighted in Section 2 of the revised paper. We also remark that we are the first to systematically investigate the construction of evaluation partners and the measurement of ZSC capability.

### Best-response Diversity (BR-Div)
We highlight that we propose a novel diversity metric, i.e., BR-Div, for population diversity in the ZSC domain. We provide justification for measuring the population diversity of a group of cooperative agents by the diversity of their best responses. The insight is that the ego agent aims at emulating the BR to any partner in the training population. **Thus an ego agent with better ZSC capability means that the ego agent emulates more BRs. To evaluate the ZSC capability, we should measure that capability of emulating a lot of BRs**, i.e., expose the ego agent to partners with a lot of BRs.

We verify the effectiveness of BR-Div in Figure 2, Figure 3, Figure 8 and Figure 9, both qualitatively and quantitatively. A simple and intuitive explanation of why BR-Div is more effective than the population diversity of partners is that two different partners may correspond to similar BRs or even the same BR [6].

---

> ### Author Response · Authors · 2023-11-17
> **Common Response (2/3)**
>
> ## Practicality
> Reviewers (6Vtg and dEYW) might be concerned that the extra human effort required for applying our evaluation workflow to evaluating ZSC methods in complex tasks diminishes the method's contribution to the community. However, we contend that **the extra effort, which consists of designing event-based rewards, policy behavior features, and training approximate best responses, is not as onerous as might be initially perceived, on the condition that the ZSC methods have been implemented in complex tasks**. The feature and reward engineering work is completed when the ZSC methods are effectively implemented, and at the very least, the self-play algorithm can train agent pairs to solve the complex tasks.
>
> **The event-based reward design is very similar to the feature and rewarding shaping, and thus one can make moderate effort to design the event-based reward**. Besides, event-based rewards have been used in lots of complex tasks, including robotics [1], large-scale games [2,3] and product management [4].
>
> Our evaluation workflow includes two kinds of approximate best responses: approximating the Nash Equilibrium (NE) of the two-player Markov game constructed with event-based rewards and approximating the best responses to earlier checkpoints of the fully trained behavior preferring agents. **The complexity of the former will not exceed training the ZSC methods, and the complexity of the latter will be lower than training a self-play agent pair** since one of the agents is fixed. The policy behavior features that are used to figure out BR-Div depend on the environment and the task at hand, which are related to the skills being tested. For simplicity, we count the occurrence of the pre-defined events counted along episodes, which hardly requires additional effort.
>
> To further explain that designing event-based rewards and policy behavior features is practical, we detailedly describe them in the following and give an example of the design used in Overcooked.
>
> ## Event-based Reward and Policy Behavior Feature Design
> Reviewers (6Vtg, dEYW and PVtL) have required more details about event-based reward and policy behavior feature design. The event-based reward space can be formulated as follows:
> $$
> \mathcal{R}=\\{r _{\bf{w}}| r _{\bf{w}}(s _{t}, \bf{a} _{t}) = r + \phi(s _{t}, \bf{a} _{t})^{T}\bf{w}, \bf{w} \in \mathbb{R}^{m}, \|\bf{w}\| _{\infty} \leq B _{\text{max}}, \sum _{i} \mathbb{1}(\bf w _{i} \neq 0) \leq C _{\text{max}}\\}
> $$
>
>
> The hyper-paramters $B_{\text{max}}$ and $C_{\text{max}}$ are used to restrict the reward space to indicate reasonable behaviors. $\phi(s _t,\bf{a} _{t})$ is the indicator of pre-defined events.
>
> The policy behavior features that are used to figure out BR-Div depend on the environment and the task at hand, which are related to the skills being tested. For simplicity, we count the occurrence of the pre-defined events, i.e., $\mathbb{E}[\sum _{t=1}^{T}\phi(s _t,\bf{a} _t) ]$, as the policy behavior feature.
>
> ### An example of Overcooked
> In Overcooked, we use $B_{\text{max}} = 20$, $C_{\text{max}}=3$ and generate up to 194 candidates and select up to 30 evaluation partners. The generated candidates are excluded if they cannot complete a delivery when cooperating with their BRs. The pre-defined events are listed as follows:
> | Events | Weights |
> | --- | --- |
> | Put an onion or a dish or a soup onto the counter  | 0  |
> | Pickup an onion or a dish or a soup from the counter | 0 |
> | Pickup an onion from the onion dispenser | -20,0,10  |
> | Pickup a dish from the dish dispenser  | -20,0,10 |
> | Pickup a soup | -20,0,5,10 |
> | Place an ingredient into the pot | -20,0,3,10 |
> | Deliver a soup | -20,0 |
> | Stay | -0.1,0,0.1 |
> | Movement | 0 |
> | Order Reward | 0.1,1 |
>
> We also provide the pseudocode for using these events to show that one can make a moderate effort to implement the event-based rewards.
> ```python=
> state = current state of the game
> action = action of the agent whose events are collected
> reward = 0
> for e_i in range(NUM_EVENTS):
>     if True == EVENT_JUDGER(e_i, state, action):
>         reward += EVENT_WEIGHTS[e_i]
> ```
> The details are also present in Appendix B.1.

---

> ### Author Response · Authors · 2023-11-17
> **Common Response (3/3)**
>
> ## Summary of the Paper Revisions
> The main updates are summarized as follows:
> 1. Page 2, Sec. 1, highlight our contribution to proposing BR-Div.
> 2. Page 3, Sec. 2, highlight the differences between our evaluation workflow and previous evaluation methods.
> 3. Page 3, Sec. 2, clarify the relations among columns in Table 1.
> 4. Page 4, Sec. 3.2, highlights the problems of previous evaluation methods.
> 5. Page 6, Sec. 4.1, add the definition of P-Div and an explanation of Figure 3.
> 6. Page 6, Sec. 4.2, add the motivation to using event-based reward shaping.
> 7. Page 7, Sec. 4.2, highlights the design of policy behavior features.
> 8. Page 7, Sec. 4.2, explain why our generated evaluation partners meet the desiderata.
> 9. Page 7-8, Sec. 5, add details about event design in Overcooked.
> 10. Page 8-9, Sec. 5, add more explanation about the experiment results.
> 11. Page 9, Sec, 6, add future work.
> 12. Figure 4, 5, 10 and 11, Table 4, add the results of HSP in Overcooked.
> 13. Page 17, Appendix, add more details about experiments.
>
> **Reference**
>
> [1] Brohan, Anthony, et al. "Do as i can, not as i say: Grounding language in robotic affordances." CoRL. 2023.
>
> [2] Kurach, Karol, et al. "Google research football: A novel reinforcement learning environment." AAAI. 2020.
>
> [3] Deheng Ye, et al. Mastering Complex Control in MOBA Games with Deep Reinforcement Learning. AAAI. 2020.
>
> [4] Shi, Zhenyu, et al. "Learning expensive coordination: An event-based deep RL approach." ICLR. 2020.
>
> [5] Chao Yu, Jiaxuan Gao, Weilin Liu, Botian Xu, Hao Tang, Jiaqi Yang, Yu Wang, and Yi Wu. "Learning zero-shot cooperation with humans, assuming humans are biased". ICLR. 2023.
>
> [6] Sarkar, Bidipta, Andy Shih, and Dorsa Sadigh. "Diverse Conventions for Human-AI Collaboration." NeurIPS. 2023.

---

### Author Response · Authors · 2023-11-20
**A Gentle Reminder for Rebuttal Consideration**

Dear Reviewers,

Thanks again for your valuable comments and suggestions. We have submitted our rebuttals to your reviews.  We explain the problems and try our best to settle the reviewers' concerns, including paper revisions and additional experimental results.

We look forward to your response and are eager to continue our discussion.

Sincerely, Authors

---

### Meta-Review · Area_Chair_F1yd · 2023-12-10

**Metareview:**

The authors suggest a new multi-step method for generating diverse agents for evaluating Zero-Shot Coordination (ZSC) agents in the Overcooked benchmark.

At the core of this method are two ideas:
1) In Overcooked,  event based rewards can be use to reward shape agents into different types of behaviours
2) Rather than optimising the pool for diversity, the authors suggest to optimise the pool for the diversity of the best responses to each of the members of the pool

On the plus side, I believe the ideas in this paper are innovative and that a lot of thought went into the development of the method, as well as the analysis / visualisations of the experimental evaluation.

On the negative side, I agree with the reviewers on some of the key criticism:
First of all, if this paper is the answer, then what is the question? If the claim is that this paper provides a "better evaluation mechanism" for ZSC, then there needs to be a way of comparing different evaluation mechanisms. For example, since ZSC is usually regarded as a proxy setting for more expensive evaluation, one option might be to show that the new proposed evaluation mechanism produces evaluations that correlate better with costly human evaluations than existing evaluation methods.

Secondly, there are a lot of rather arbitrary / hand coded decisions in the paper. For example, the way that “sabotage behaviour” is removed seems very ad-hoc compared with more principled approaches in the literature. So a lower bar to improve the paper is to show that the evaluation metric is robust in all of these somewhat arbitrary decisions. Clearly, if the specific reward design space or the threshold chosen for excluding sabotage changed the evaluation, this is bad news for the generality of the metric.

Thirdly – the optimal ZSC solution according to this evaluation is to play a best response to exactly this pool. While the pool is not accessible at training time, a training algorithm can simply construct the pool as a first step and then train a best response agent as a second step. This doesn't seem like a desired outcome.

Lastly, the paper finds that all existing methods do similarly poorly when evaluated according to this metric on Overcooked and concludes that the methods are faulty, but the metric is good. An alternative interpretation is that the combination of evaluation method and problem setting is flawed. This seems likely, since various pieces of work have shown that Overcooked barely contains real coordination challenges.

One option for this paper might be to break out the paper into two steps:
1) Show that it is possible to achieve more meaningfully diverse sets of policies across a range of different environments. Diversity has been recognised as an interesting challenge by the community, so there are prior results to compare to.
2) Suggest a better problem definition / evaluation protocol for ZSC based on (1). This would likely need to be supported by empirical data or theoretical insights which address points 1- 4 above.

**Justification For Why Not Higher Score:**

See weaknesses listed above.

**Justification For Why Not Lower Score:**

NA

---

### Decision · Program_Chairs · 2024-01-16

Reject